# Mitigating Time Discretization Challenges with WeatherODE: A Sandwich Physics-Driven Neural ODE for Weather Forecasting

## Abstract

In the field of weather forecasting, traditional models often grapple with discretization errors and time-dependent source discrepancies, which limit their predictive performance. In this paper, we present WeatherODE, a novel one-stage, physics-driven ordinary differential equation (ODE) model designed to enhance weather forecasting accuracy. By leveraging wave equation theory and integrating a time-dependent source model, WeatherODE effectively addresses the challenges associated with time-discretization error and dynamic atmospheric processes. Moreover, we design a CNN-ViT-CNN sandwich structure, facilitating efficient learning dynamics tailored for distinct yet interrelated tasks with varying optimization biases in advection equation estimation. Through rigorous experiments, WeatherODE demonstrates superior performance in both global and regional weather forecasting tasks, outperforming recent state-of-the-art approaches by significant margins of over 40.0% and 31.8% in root mean square error (RMSE), respectively. The source code is available at https://anonymous.4open.science/r/WeatherODE-5C13/.

## 1 Introduction

Weather forecasting is a cornerstone of modern society, affecting key industries like agriculture, transportation, and disaster management (Coiffier, 2011). Accurate predictions help mitigate the effects of extreme weather and optimize economic operations. Recent advancements in high-performance computing have significantly boosted the accuracy and speed of numerical weather forecasting (NWP) (Bauer et al., 2015; Lorenc, 1986; Kimura, 2002).

The swift advancement of deep learning has opened up a promising avenue for weather forecasting (Weyn et al., 2019; Scher & Messori, 2019; Rasp et al., 2020a; Weyn et al., 2021; Bi et al., 2023; Pathak et al., 2022; Hu et al., 2023). However, the existing weather forecasting models based on deep learning often fail to fully account for the key physical mechanisms governing small-scale, complex nonlinear atmospheric phenomena, such as turbulence, convection, and airflow. These dynamic processes are crucial to the formation and evolution of weather systems, but most models focus on learning statistical correlations from historical data instead of explicitly extracting or integrating these physical dynamics. Furthermore, these models typically rely on fixed time intervals (e.g., every 6 hours) for predictions, limiting their applicability to varying temporal scales. Consequently, separate models are often required for different forecast periods (Bi et al., 2023), which constrains flexibility and reduces generalization.

Another line of research utilizes neural ODEs (Chen et al., 2018) that incorporate partial differential equations to guide the physical dynamics of weather forecasting. Among these methods, the advection continuity equation stands out as a key equation governing many weather indicators:

$$\frac{\partial u}{\partial t} + \underbrace{v \cdot \nabla u + u \nabla \cdot v}_{\text{Advection}} = \underbrace{s}_{\text{Source}}, \tag{1}$$

where $u$ represents a atmospheric variable evolving over space and time, driven by the flow velocity $v$ and the source term $s$. A recent study, ClimODE (Verma et al., 2024), effectively employs this

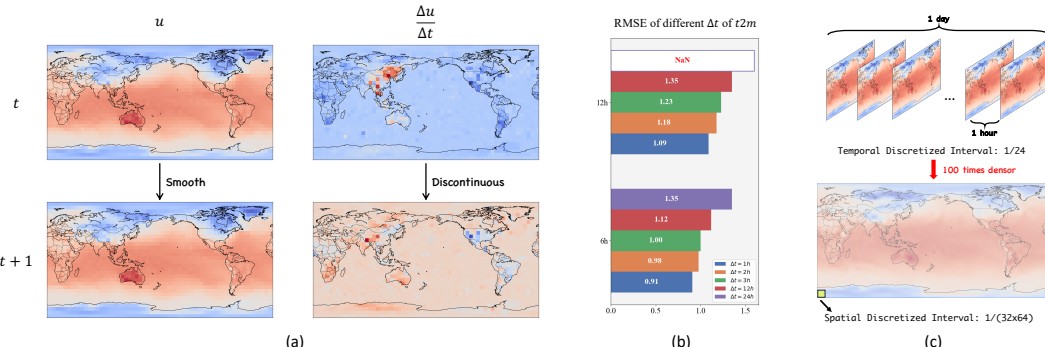

Figure 1: (a) Comparison of two-meter temperature ($t2m$) and its discrete-time derivative over a 1-hour interval. While the temperature evolves continuously, the discrete-time derivative exhibits discontinuities, leading to discretization errors. (b) Latitude-weighted RMSE for $t2m$ using models trained with different time intervals ($\Delta t$) for estimating initial velocity. Larger $\Delta t$ values result in worse performance and can even lead to numerical instability (NaN). See Table 12 for full results. (c) Comparison of temporal and spatial discretization intervals in the 5.625° ERA5 dataset. The spatial discretization is 100 times denser than the temporal discretization.

equation and achieves state-of-the-art performance. However, there are several inherent challenges when solving such equations using neural ODEs. Firstly, the accurate estimation of the initial velocity is crucial to the weather forecasting performance. Unfortunately, current methods typically rely on time discretization to estimate atmospheric time gradients for velocity calculation and cannot achieve satisfying accuracy. In particular, we face a constraint due to a 1-hour discretization limit imposed by the temporal resolution of the ERA5 dataset, which is usually chosen for training of deep models including most global weather forecasting models. As shown in Figure 1a, it is evident that velocity estimation is far from continuous, despite the observed variable being relatively smooth and continuous. Furthermore, we demonstrate in Figure 1b that using larger discretization intervals for velocity estimation would significantly hinder our forecasting performance. This indicates that 1-hour estimates can introduce significant errors. On the other hand, we note that coarse calculations from 5.625° ERA5 data (Rasp et al., 2020b) reveal a temporal resolution of 1/24 and a spatial resolution of 1/(32×64), resulting in the spatial domain nearly 100 times denser, which can help to reduce errors from temporal discretization (Figure 1c). Secondly, to better solve the advection equation, we need to consider three key components carefully, including the initial velocity estimation, solving the advection equation itself, and the error term arising from deviations in reality. Due to their physical nature, they call for different modeling. For example, global and long-term interactions govern the advection process, while local and short-term interactions dictate the velocity estimation and equation's overall deviations. Lastly, the source term should be modeled as time-dependent for better estimation.

To address these challenges, we propose WeatherODE, a one-stage, physics-driven ODE model for weather forecasting. It leverages the wave equation, widely used in atmospheric simulations, to improve the estimation of initial velocity using more precise spatial information $\nabla u$. This approach reduces the time-discretization errors introduced by using $\frac{\Delta u}{\Delta t}$. Additionally, we introduce a time-dependent source model that effectively captures the evolving dynamics of the source term. Furthermore, we have meticulously crafted the model architecture to seamlessly integrate local feature extraction with global context modeling, promoting efficient learning dynamics tailored for three tasks in advection equation estimation. Our contributions can be summarized as follows:

- We conduct thorough experiments to identify and demonstrate the issues of discretization error and time-dependent source error, both of which significantly hinder the performance of current physics-informed weather forecasting models.

- We propose WeatherODE, a one-stage, physics-driven ODE model for weather forecasting that utilizes wave equation theory and a time-dependent source model to address the identified challenges. To solve the advection equation more accurately, we conduct a comprehensive analysis of the architectural design of the CNN-ViT-CNN sandwich structure,

facilitating efficient learning dynamics tailored for distinct yet interrelated tasks with varying optimization biases.

- WeatherODE demonstrates impressive performance in both global and regional weather forecasting tasks, significantly surpassing the recent state-of-the-art methods by margins of 40.0% and 31.8% in RMSE, respectively.

## 2 RELATED WORKS

The most advanced weather forecasting techniques predominantly rely on Numerical Weather Prediction (NWP) (Lorenc, 1986; Kimura, 2002), which employs a set of equations solved on supercomputers to model and predict the atmosphere. While NWP has achieved promising results, it is resource-intensive, requiring significant computational power and domain expertise to define the appropriate physical equations.

Deep learning-based weather forecasting adopts a data-driven approach to learning the spatio-temporal relationships between atmospheric variables. These methods can be broadly classified into Graph Neural Networks (GNN) and Transformer-based methods. GNN-based methods (Lam et al., 2022; Keisler, 2022) treat the Earth as a graph and use graph neural networks to predict weather patterns. Transformer-based approaches have shown significant success in weather forecasting due to their scalability (Chen et al., 2023b;a; Han et al., 2024; Vaswani, 2017). For example, Pangu (Bi et al., 2023) employs a 3D Swin Transformer (Liu et al., 2021) and an autoregressive model to accelerate inference. Fengwu (Chen et al., 2023a) models atmospheric variables as separate modalities and uses a replay buffer for optimization, with Fengwu-GHR (Han et al., 2024) subsequently extending the approach to higher-resolution data. Additionally, ClimaX (Nguyen et al., 2023) and Aurora (Bodnar et al., 2024) introduce a pretraining-finetuning framework, where models are first pretrained on physics-simulated data and then finetuned on real-world data. However, these models frequently neglect the fundamental physical dynamics of the atmosphere and are limited to providing fixed lead time for each prediction.

Physics-driven methods, which integrate physical constraints in the form of partial differential equations (PDEs) (Evans, 2022) into neural networks, have gained increasing attention in recent years (Cai et al., 2021; Li et al., 2024b). In weather forecasting, DeepPhysiNet (Li et al., 2024a) incorporates physical laws into the loss function, marking an initial attempt to combine neural networks with PDEs. ClimODE (Verma et al., 2024) advances further by leveraging the continuity equation to express the weather forecasting process as a full PDE system modeled using neural ODEs (Chen et al., 2018). NeuralGCM (Kochkov et al., 2024) incorporates more physical constraints and designs neural networks to function as a dynamic core. However, it is complex and difficult to modify, as it operates with over a dozen ODE functions similar to the NWP method. In contrast, our proposed WeatherODE offers a more straightforward and efficient foundation for ongoing improvements.

## 3 METHOD

In this section, we first introduce the overall ODE modeling framework for weather forecasting in Section 3.1. We then describe the specific designs of the Velocity Model, Advection ODE, and Source Model in Section 3.2, Section 3.3, and Section 3.4, respectively. We present the overarching design choices for our CNN-ViT-CNN sandwich structure in Section 3.5. Finally, we end up with the multi-task learning strategy in Section 3.6.

### 3.1 ODE FRAMEWORK FOR WEATHER DYNAMICS

We can model the atmosphere as a spatio-temporal process $\mathbf{u}(x, y, t) = (u_1(x, y, t), \ldots, u_K(x, y, t)) \in \mathbb{R}^K$, where $K$ represents the number of distinct atmospheric variables $u_k(x, y, t) \in \mathbb{R}$, evolving over continuous time $t$ and spatial coordinates $(x, y) \in [0, H] \times [0, W]$, $H$ and $W$ are the height and width, respectively. Each quantity or atmospheric variable is driven by a velocity field $v_k(x, y, t) \in \mathbb{R}^{2K}$ and influenced by a source term $s_k(x, y, t) \in \mathbb{R}^K$. For simplicity, we first omit the index $k$ since all quantities are treated equally, and then drop the spatial coordinates $(x, y)$ to focus on the time evolution. The time derivative is

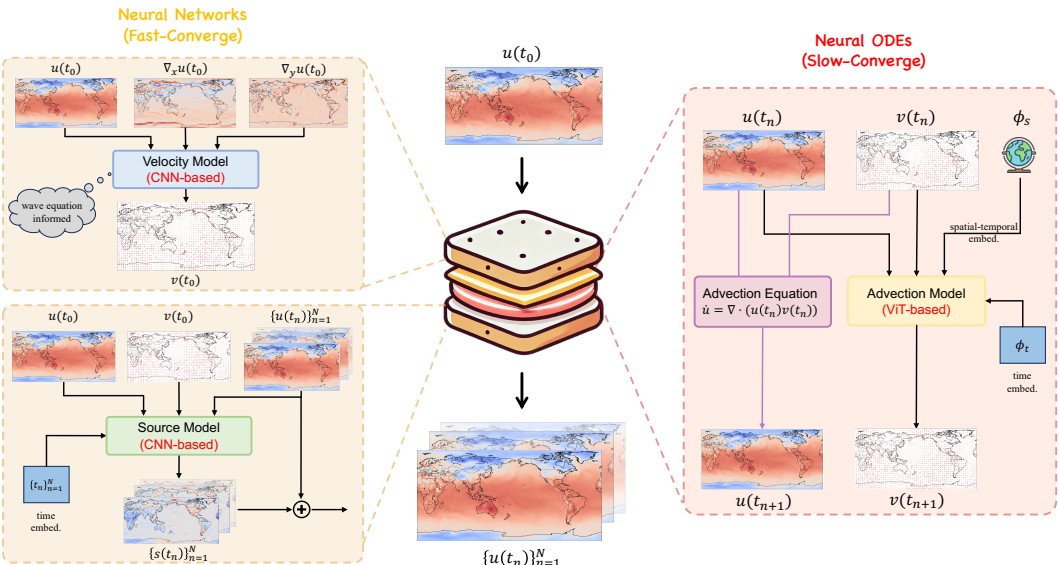

Figure 2: Overall architecture of WeatherODE. WeatherODE adopts a sandwich-like structure for atmosphere modeling. The top and bottom parts use fast-converging neural networks (CNN-based) to estimate the initial velocity and source term, while the central layer employs a slower-converging neural ODE (ViT-based) to model the atmospheric advection process. This design ensures stability when training the neural ODE to solve the numerical solution. More analyses are in Section 3.5 and Section 5.3.

denoted as $\dot{u}$ (i.e., $\frac{\partial u}{\partial t}$), while spatial variation is captured through the gradient $\nabla u$ (i.e., $\frac{\partial u}{\partial x}$ and $\frac{\partial u}{\partial y}$). Based on Equation 1, we hypothesize that the atmospheric system follows the subsequent partial differential equation:

$$\dot{u}(t) = \underbrace{-v(t) \cdot \nabla u(t) - u(t)\nabla \cdot v(t)}_{\text{Advection}} + s(t).$$
(2)

Using the Method of Lines, we can express Equation 2 as a continuous first-order ODE system (Verma et al., 2024). In practice, the system is discretized into $N$ time steps $\{t_1, \ldots, t_N\}$, which allows us to leverage data from multiple future points to supervise the ODE in intermediate steps and apply numerical solvers like the Euler method (Biswas et al., 2013). This results in the following discretized form:

$$\begin{bmatrix} u(t_{n+1}) \\ v(t_{n+1}) \end{bmatrix} = \underbrace{\begin{bmatrix} u(t_n) \\ v(t_n) \end{bmatrix}}_{\text{Initial Velocity } v(t_0)} + \Delta t \underbrace{\begin{bmatrix} -\nabla \cdot (u(t_n)v(t_n)) \\ \dot{v}(t_n) \end{bmatrix}}_{\text{Advection ODE}} + \underbrace{\begin{bmatrix} s(t_n) \\ 0 \end{bmatrix}}_{\text{Source Term}}.$$
(3)

To solve this ODE system, three unknowns need to be estimated: $v(t_0)$, $\dot{v}(t_n)$, and $s(t_n)$. As shown in Figure 2, the proposed WeatherODE uses neural networks to model $v(t_0)$ and $s(t_n)$, and a neural ODE to model $\dot{v}(t_n)$, which will be discussed in the following sections.

## 3.2 VELOCITY MODEL

Modeling the initial velocity $v(t_0)$ is crucial for ensuring the stability of the ODE solution. ClimODE (Verma et al., 2024) estimates the initial velocity by first calculating the discrete-time derivative $\frac{\Delta u}{\Delta t}$ from several past time points. However, using the discrete approximation $\frac{\Delta u}{\Delta t}$ introduces large numerical errors, especially when $\Delta t$ is not small enough. This approach struggles to capture smooth variations, resulting in significant deviations from the true continuous derivatives. Moreover, it involves a two-stage process where a separate model must first be trained to estimate all initial values $v(t_0)$ before proceeding with the ODE solution.

Therefore, based on the following assumptions, we introduce the wave equation to leverage more precise spatial information for estimating the initial velocity.

**Incompressibility**: In this study, we assume that the fluid (air) behaves as incompressible. This implies that variations in pressure do not significantly influence the density of the fluid. This assumption is generally valid for large-scale weather phenomena; however, it may not be applicable to smaller, localized events.

**Linearization**: The governing equations of atmospheric dynamics can be linearized around a mean state, permitting the examination of small perturbations. This approach simplifies the mathematical framework and facilitates the superposition of solutions.

Given these assumptions, we can utilize the wave equation (Evans, 2022), commonly employed in atmospheric simulations, to enhance the estimation of the initial velocity based on the available spatial information, as outlined below:

$$\frac{\partial^2 u}{\partial t^2} = c^2 \left( \frac{\partial^2 u}{\partial x^2} + \frac{\partial^2 u}{\partial y^2} \right). \tag{4}$$

This allows the first derivative with respect to time to be expressed as:

$$\frac{\partial u}{\partial t} = \int c^2 \left( \frac{\partial^2 u}{\partial x^2} + \frac{\partial^2 u}{\partial y^2} \right) dt. \tag{5}$$

Thus, $\frac{\partial u}{\partial t}$ can be accurately computed as a function of the spatial derivatives $\frac{\partial u}{\partial x}$ and $\frac{\partial u}{\partial y}$, avoiding additional numerical errors. We model $v(t_0)$ using a CNN-based neural network $f_v(\cdot)$:

$$v(t_0) = f_v(u(t_0), \nabla u(t_0)).$$

However, there is no free lunch, as we must also consider the discretization errors we introduce in the spatial domains. Coarse estimations based on 5.625° ERA5 data (Rasp et al., 2020b) suggest a temporal resolution of $1/24$ and a spatial resolution of $1/(32*64)$, indicating that the spatial domain is nearly 100 times denser than the temporal domain. This disparity allows our approach to deliver a more precise and stable estimation of the initial velocity, which is vital for accurately solving the ODE system.

### 3.3 ADVECTION ODE

In the discretized ODE system in Equation 3, the term $\dot{u}(t_n)$ can be computed from the current values of $u(t_n)$ and $v(t_n)$ using the advection equation. For $\dot{v}(t_n)$, we design an advection model:

$$\dot{v}(t_n) = f_\theta(u(t_n), \nabla u(t_n), v(t_n), (\phi_s, \phi_t)),$$

where $(\phi_s, \phi_t)$ represent the spatial-temporal embeddings and details can be found in Appendix C.2.

The design of the advection model $f_\theta$ is crucial for ensuring the stability of the numerical solution, as it takes the output from the velocity model as input. We argue that $f_\theta$ should converge more slowly than the CNN-based velocity model, because the initial estimates of $v(t_0)$ from the velocity model are likely to be inaccurate. If $f_\theta$ converges too quickly based on early, imprecise values, it could cause the numerical solution to become unstable, potentially leading to failure during optimization.

To address this, $f_\theta$ is designed with a Vision Transformer (ViT) (Dosovitskiy et al., 2021) as the primary network, complemented by a linear term. The ViT, with its inherently slower convergence relative to CNNs, provides strong global modeling capabilities, while the linear term contributes to stabilizing the training process by promoting smoother convergence (Linot et al., 2023). A detailed analysis of how different architectural choices impact training stability is available in Section 5.3.

### 3.4 SOURCE MODEL

To capture the energy gains and losses within the ODE system, we introduce a neural network to model the source term. Rather than incorporating the source term directly within the Advection ODE, we model it separately using the output of the Advection ODE $\{u(t_n)\}_{n=1}^{N}$ to predict the corresponding source terms $\{s(t_n)\}_{n=1}^{N}$. This approach mitigates the numerical errors that would arise

from modeling $s(t_i)$ within the ODE solver, as these errors would propagate through the solution. The source model $f_s(\cdot)$ is formulated as follows:

$$\{s(t_n)\}_{n=1}^N = f_s(\{u(t_n)\}_{n=1}^N, u(t_0), v(t_0), \phi_s, \{t_n\}_{n=1}^N).$$

This model is supervised using the predicted values $\{u(t_n)\}_{n=1}^N$, the spatial embedding $\phi_s$, the sequence of time points $\{t_n\}_{n=1}^N$, and the initial conditions $u(t_0)$ and $v(t_0)$. Rather than assuming the source term is independent at each time step, the model captures its temporal evolution, considering dependencies on both past and future values. The architecture of the source model is based on a 3D CNN, with further architectural details discussed in Section 5.2.

### 3.5 SANDWICH STRUCTURE DESIGN FOR SOLVING ADVECTION EQUATION

The hybrid CNN-ViT-CNN architecture optimally combines local feature extraction and global context modeling, enabling efficient learning dynamics suited for distinct yet interconnected tasks in the advection equation estimation.

The sandwich design of our neural ODE model, comprising a CNN for fast-converging tasks (velocity estimation and source term modeling) and a ViT for slower-converging tasks (advection equation modeling), leverages the strengths of different architectures tailored to specific learning tasks. CNNs excel at local feature extraction and are particularly suited for tasks requiring rapid convergence, such as deriving initial conditions and identifying impacts from source terms with high spatial correlation. In contrast, Vision Transformers (ViTs) utilize attention mechanisms that capture global context and relationships, making them better suited for tasks with more complex interactions, such as solving the advection equation, where the dynamics often involve long-range dependencies. From a theoretical standpoint, the effectiveness of this hybrid architecture can be framed through the lens of inductive biases: the CNN's ability to model locality and translation invariance complements the ViT's ability to model global interactions and dependencies, resulting in a more robust solution strategy for the coupled problem. Moreover, such sandwich design choice is also related to the robustness of training as we discuss in Section 5.3.

### 3.6 MULTI-TASK LEARNING

Previous methods often train models using only the target leading time $u(t_N)$ as the supervision signal, ignoring the valuable information contained in intermediate states $\{u(t_n)\}_{n=1}^{N-1}$. Here, we adopt a multi-task learning strategy and leverage the continuous nature of neural ODE to predict the state at every intermediate time step $\{u(t_n)\}_{n=1}^N$, minimizing the latitude-weighted RMSE between the predicted values $u(t_n)$ and the ground truth $\tilde{u}(t_n)$. The loss function is defined as:

$$\mathcal{L} = \frac{1}{N \times K \times H \times W} \sum_{n=1}^N \sum_{k=1}^K \sum_{h=1}^H \sum_{w=1}^W \alpha(h) \left( \tilde{u}_{k,h,w}(t_n) - u_{k,h,w}(t_n) \right)^2, \tag{6}$$

where $\alpha(h)$ is the latitude weighting factor that accounts for the varying grid cell areas on a spherical Earth, as cells near the equator cover larger areas than those near the poles. For more details on the weighting factor, refer to Appendix B.

By leveraging the multi-task learning strategy, the ODE system can exploit information across different time points, helping the model filter out errors arising from advection assumptions and neural network predictions. This allows us to train a single model with a lead time of $N$ that can be used for inference at any time step up to $N$, enhancing both efficiency and generalization.

## 4 EXPERIMENTS

In this section, we evaluate the proposed WeatherODE by forecasting the weather at a future time $u(t+\Delta t)$ based on the conditions at a given time $t$, where $\Delta t$ (measured in hours) represents the lead time. The experimental setups are detailed in Section 4.1, while the results for global and regional weather forecasting are presented in Section 4.2 and Section 4.3, respectively.

### 4.1 EXPERIMENTAL SETUPS

**Dataset.** We utilize the preprocessed ERA5 dataset from WeatherBench (Rasp et al., 2020b), which has 5.625° resolution ($32 \times 64$ grid points) and temporal resolution of 1 hour. Our input data includes $K = 48$ variables: 6 atmospheric variables at 7 pressure levels, 3 surface variables, and 3 constant fields. To evaluate the performance of WeatherODE, following the benchmark work in Verma et al. (2024), we focus on five target variables: geopotential at 500 hPa ($z500$), temperature at 850 hPa ($t850$), temperature at 2 meters ($t2m$), and zonal wind speeds at 10 meters ($u10$ and $v10$). We use the data from 1979 to 2015 as the training set, 2016 as the validation set, and 2017 to 2018 as the test set. More details are available in Appendix A.

**Metric.** In line with previous works, we evaluate all methods using latitude-weighted root mean squared error (RMSE) and latitude-weighted anomaly correlation coefficient (ACC):

$$\text{RMSE} = \frac{1}{K} \sum_{k=1}^{K} \sqrt{\frac{1}{HW} \sum_{h=1}^{H} \sum_{w=1}^{W} \alpha(h) \left(\tilde{u}_{k,h,w} - u_{k,h,w}\right)^2}, \tag{7}$$

$$\text{ACC} = \frac{\sum_{k,h,w} \tilde{u}'_{k,h,w} u'_{k,h,w}}{\sqrt{\sum_{k,h,w} \alpha(h)(\tilde{u}'_{k,h,w})^2 \sum_{k,h,w} \alpha(h)(u'_{k,h,w})^2}},$$

where $\alpha(h)$ is the same latitude weighting factor as used in the training process; $\tilde{u}' = \tilde{u} - C$ and $u' = u - C$ are computed against the climatology $C = \frac{1}{K} \sum_k \tilde{u}_k$, which is the temporal mean of the ground truth data over the entire test set. More details are available in Appendix B.

**Baselines.** We compare WeatherODE with several representative methods from recent literature, including ClimaX (Nguyen et al., 2023), FourCastNet (FCN) (Pathak et al., 2022), ClimODE (Verma et al., 2024), and the Integrated Forecasting System (IFS) (ECMWF, 2023). Specifically, ClimaX is a pre-trained framework capable of learning from heterogeneous datasets that span different variables, spatial and temporal scales, and physical bases. FCN uses Adaptive Fourier Neural Operators to provide fast, high-resolution global weather forecasts. ClimODE is a physics-informed neural ODE model that incorporates key physical principles. IFS is the most advanced global physics simulation model of the European Center for Medium-Range Weather Forecasting (ECMWF).

**Implementation details.** The architecture of our velocity model is based on ResNet2D (He et al., 2016), the ODE is based on ViT (Dosovitskiy et al., 2021), and the source model is based on ResNet3D. We optimize the model using the Adam optimizer. Detailed discussions on the model architectures, specific parameter settings, and learning rate schedules are available in Appendix C.

### 4.2 GLOBAL WEATHER FORECASTING

Table 1 presents the global weather forecasting performance of WeatherODE and other baseline models at $\Delta t = \{6, 12, 18, 24\}$ hours. We report the results from the original ClimaX paper, where the model was pre-trained on the CMIP6 dataset (Eyring et al., 2016) and then fine-tuned on ERA5 dataset. Despite training solely on the ERA5 dataset, WeatherODE gains a 10% improvement over ClimaX. Besides, WeatherODE surpasses ClimODE with a substantial improvement over 40%, clearly demonstrating that we have effectively overcome the major challenges inherent in physics-driven weather forecasting models. Furthermore, WeatherODE achieves performance on par with the IFS, which serves as the benchmark in the industry.

### 4.3 REGIONAL WEATHER FORECASTING

Global forecasting is not always feasible when only regional data is available. Therefore, we evaluate WeatherODE with other baselines for regional forecasting of relevant variables in North America, South America, and Australia, focusing on predicting future weather in each region based on its current conditions. The latitude boundaries for these regions are detailed in the Appendix D. As shown in Table 2, WeatherODE consistently achieves strong predictive performance across nearly all variables in each region, surpassing ClimaX and ClimODE by 59.7% and 31.8%, respectively.

| Variable | Hours | RMSE ↓ | | | | | | ACC ↑ | | | | | |
|---|---|---|---|---|---|---|---|---|---|---|---|---|---|
| | | ClimaX [†] (2023) | FCN (2022) | IFS (2023) | ClimODE (2024) | WeatherODE (**Ours**) | WeatherODE* (**Ours**) | ClimaX [†] (2023) | FCN (2022) | IFS (2023) | ClimODE (2024) | WeatherODE (**Ours**) | WeatherODE* (**Ours**) |
| z500 | 6 | 62.7 | 149.4 | 26.9 | 102.9 | 54.0 | 56.3 | 1.00 | 0.99 | 1.00 | 0.99 | 1.00 | 1.00 |
| | 12 | 81.9 | 217.8 | (N/A) | 134.8 | 80.0 | 73.3 | 1.00 | 0.99 | (N/A) | 0.99 | 1.00 | 1.00 |
| | 18 | 88.9 | 275.0 | (N/A) | 162.7 | 96.3 | 91.9 | 1.00 | 0.99 | (N/A) | 0.98 | 1.00 | 1.00 |
| | 24 | 96.2 | 333.0 | 51.0 | 193.4 | 114.5 | 114.5 | 1.00 | 0.99 | 1.00 | 0.98 | 1.00 | 1.00 |
| t850 | 6 | 0.88 | 1.18 | 0.69 | 1.16 | 0.73 | 0.76 | 0.98 | 0.99 | 0.99 | 0.97 | 0.99 | 0.99 |
| | 12 | 1.09 | 1.47 | (N/A) | 1.32 | 0.87 | 0.88 | 0.98 | 0.99 | (N/A) | 0.96 | 0.98 | 0.98 |
| | 18 | 1.10 | 1.65 | (N/A) | 1.47 | 0.95 | 0.95 | 0.98 | 0.99 | (N/A) | 0.96 | 0.98 | 0.98 |
| | 24 | 1.11 | 1.83 | 0.87 | 1.55 | 1.04 | 1.04 | 0.98 | 0.99 | 0.99 | 0.95 | 0.98 | 0.98 |
| t2m | 6 | 0.95 | 1.28 | 0.97 | 1.21 | 0.74 | 0.78 | 0.98 | 0.99 | 0.99 | 0.97 | 0.99 | 0.99 |
| | 12 | 1.24 | 1.48 | (N/A) | 1.45 | 0.88 | 0.89 | 0.97 | 0.99 | (N/A) | 0.96 | 0.99 | 0.98 |
| | 18 | 1.19 | 1.61 | (N/A) | 1.43 | 0.95 | 0.95 | 0.97 | 0.99 | (N/A) | 0.96 | 0.98 | 0.98 |
| | 24 | 1.10 | 1.68 | 1.02 | 1.40 | 0.98 | 0.98 | 0.98 | 0.99 | 0.99 | 0.96 | 0.98 | 0.98 |
| u10 | 6 | 1.08 | 1.47 | 0.80 | 1.41 | 0.84 | 0.88 | 0.97 | 0.95 | 0.98 | 0.91 | 0.98 | 0.98 |
| | 12 | 1.23 | 1.89 | (N/A) | 1.81 | 1.00 | 1.00 | 0.95 | 0.93 | (N/A) | 0.89 | 0.97 | 0.97 |
| | 18 | 1.27 | 2.05 | (N/A) | 1.97 | 1.12 | 1.13 | 0.95 | 0.91 | (N/A) | 0.88 | 0.96 | 0.96 |
| | 24 | 1.41 | 2.33 | 1.11 | 2.01 | 1.26 | 1.26 | 0.94 | 0.89 | 0.97 | 0.87 | 0.95 | 0.95 |
| v10 | 6 | (N/A) | 1.54 | 0.94 | 1.53 | 0.87 | 0.90 | (N/A) | 0.94 | 0.98 | 0.92 | 0.98 | 0.98 |
| | 12 | (N/A) | 1.81 | (N/A) | 1.81 | 1.04 | 1.04 | (N/A) | 0.91 | (N/A) | 0.89 | 0.97 | 0.97 |
| | 18 | (N/A) | 2.11 | (N/A) | 1.96 | 1.15 | 1.16 | (N/A) | 0.86 | (N/A) | 0.88 | 0.96 | 0.96 |
| | 24 | (N/A) | 2.39 | 1.33 | 2.04 | 1.29 | 1.29 | (N/A) | 0.83 | 0.97 | 0.86 | 0.95 | 0.95 |

[†] For 6h and 24h, we report results from the original ClimaX paper; 12h and 18h results are obtained using their official pre-trained model and code[1].

[*] Indicates a 24-hour model used for inference across all lead times.

Table 1: Latitude-weighted RMSE and ACC comparison with baseline models for various target variables across different lead times on global weather forecasting.

| Variable | Hours | North-America | | | | South-America | | | | Australia | | | |
|---|---|---|---|---|---|---|---|---|---|---|---|---|---|
| | | ClimaX[†] (2023) | ClimODE (2024) | WeatherODE (**Ours**) | WeatherODE* (**Ours**) | ClimaX[†] (2023) | ClimODE (2024) | WeatherODE (**Ours**) | WeatherODE* (**Ours**) | ClimaX[†] (2023) | ClimODE (2024) | WeatherODE (**Ours**) | WeatherODE* (**Ours**) |
| z500 | 6 | 273.4 | 134.5 | 91.2 | 97.3 | 205.4 | 107.7 | 62.3 | 68.9 | 190.2 | 103.8 | 62.7 | 58.4 |
| | 12 | 329.5 | 225.0 | 147.4 | 158.7 | 220.2 | 169.4 | 97.7 | 100.0 | 184.7 | 170.7 | 79.2 | 77.7 |
| | 18 | 543.0 | 307.7 | 218.9 | 233.5 | 269.2 | 237.8 | 137.5 | 141.2 | 222.2 | 211.1 | 103.5 | 102.7 |
| | 24 | 494.8 | 390.1 | 314.5 | 314.5 | 301.8 | 292.0 | 183.1 | 183.1 | 324.9 | 308.2 | 125.1 | 125.1 |
| t850 | 6 | 1.62 | 1.28 | 0.88 | 0.94 | 1.38 | 0.97 | 0.73 | 0.77 | 1.19 | 1.05 | 0.65 | 0.64 |
| | 12 | 1.86 | 1.81 | 1.09 | 1.15 | 1.62 | 1.25 | 0.91 | 0.92 | 1.30 | 1.20 | 0.76 | 0.76 |
| | 18 | 2.75 | 2.03 | 1.28 | 1.35 | 1.79 | 1.43 | 1.06 | 1.07 | 1.39 | 1.33 | 0.87 | 0.86 |
| | 24 | 2.27 | 2.23 | 1.57 | 1.57 | 1.97 | 1.65 | 1.25 | 1.25 | 1.92 | 1.63 | 0.97 | 0.97 |
| t2m | 6 | 1.75 | 1.61 | 0.66 | 0.71 | 1.85 | 1.33 | 0.80 | 0.86 | 1.57 | 0.80 | 0.73 | 0.71 |
| | 12 | 1.87 | 2.13 | 0.78 | 0.84 | 2.08 | 1.04 | 0.96 | 0.98 | 1.57 | 1.10 | 0.81 | 0.81 |
| | 18 | 2.27 | 1.96 | 0.86 | 0.93 | 2.15 | 0.98 | 1.07 | 1.08 | 1.72 | 1.23 | 0.89 | 0.88 |
| | 24 | 1.93 | 2.15 | 0.99 | 0.99 | 2.23 | 1.17 | 1.17 | 1.17 | 2.15 | 1.25 | 0.93 | 0.93 |
| u10 | 6 | 1.74 | 1.54 | 1.05 | 1.09 | 1.27 | 1.25 | 0.83 | 0.87 | 1.40 | 1.35 | 1.02 | 1.04 |
| | 12 | 2.24 | 2.01 | 1.37 | 1.42 | 1.57 | 1.49 | 1.05 | 1.03 | 1.77 | 1.78 | 1.24 | 1.27 |
| | 18 | 3.24 | 2.17 | 1.77 | 1.81 | 1.83 | 1.81 | 1.19 | 1.20 | 2.03 | 1.96 | 1.39 | 1.45 |
| | 24 | 3.14 | 2.34 | 2.22 | 2.22 | 2.04 | 2.08 | 1.39 | 1.39 | 2.64 | 2.33 | 1.62 | 1.62 |
| v10 | 6 | 1.83 | 1.67 | 1.12 | 1.16 | 1.31 | 1.30 | 0.89 | 0.92 | 1.47 | 1.44 | 1.09 | 1.10 |
| | 12 | 2.43 | 2.03 | 1.52 | 1.57 | 1.64 | 1.71 | 1.11 | 1.10 | 1.79 | 1.87 | 1.28 | 1.32 |
| | 18 | 3.52 | 2.31 | 2.00 | 2.05 | 1.90 | 2.07 | 1.26 | 1.28 | 2.33 | 2.23 | 1.41 | 1.48 |
| | 24 | 3.39 | 2.50 | 2.56 | 2.56 | 2.14 | 2.43 | 1.49 | 1.49 | 2.58 | 2.53 | 1.64 | 1.64 |

[†] The number is cited from ClimODE (Verma et al., 2024).

Table 2: Latitude-weighted RMSE comparison with baseline models for various target variables across different lead times on regional weather forecasting.

This underscores the strong ability of WeatherODE to model weather patterns effectively in data-scarce scenarios.

## 4.4 FLEXIBLE INFERENCE WITH A SINGLE 24-HOUR MODEL

Many deep learning-based methods treat predictions for different lead times as separate tasks, requiring a distinct model for each lead time. Some approaches attempt to use short-range models with rolling strategies (Bi et al., 2023; Chen et al., 2023a), but they still face the challenge of error accumulation. In contrast, by modeling the atmosphere as a physics-driven continuous process and

[1] https://github.com/microsoft/ClimaX

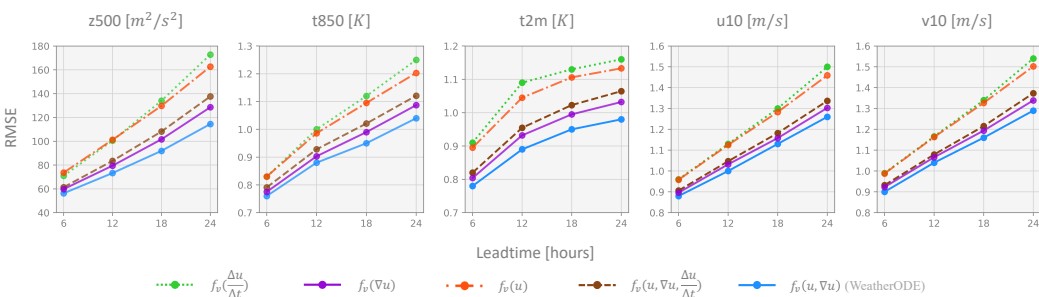

Figure 3: RMSE comparison for different input configurations of the velocity model.

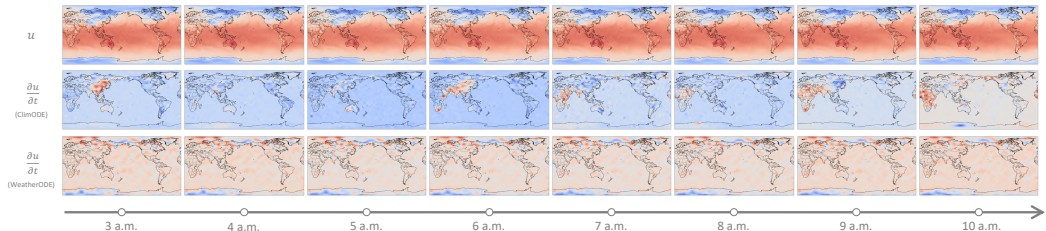

Figure 4: Visualization of the 2-meter temperature $u$ on January 1, 2017, from 3 a.m. to 10 a.m., with the estimated $\frac{\partial u}{\partial t}$ from ClimODE and WeatherODE. WeatherODE provides smoother, more continuous estimates of $\frac{\partial u}{\partial t}$, closely matching $u$, while ClimODE shows abrupt changes.

designing a time-dependent source network to account for errors at each time step, WeatherODE can capture information across all intermediate time points. As shown in Table 1 and Table 2, WeatherODE* (a 24-hour model of WeatherODE used for inference across all lead times) demonstrates its effectiveness for any hour within that period. The results show that WeatherODE* achieves performance comparable to WeatherODE across most variables and even exceeds WeatherODE in certain cases (e.g., $z500$). This highlights the effectiveness of our physics-driven ODE model in filtering out accumulated errors.

## 5 ABLATION STUDIES

### 5.1 EFFECTIVENESS OF WAVE EQUATION-INFORMED ESTIMATION

To validate the superiority of the wave equation-informed estimation over the discrete-time derivative, we conduct five experiments of the velocity model to estimate the initial velocity: (1) $f_v(\frac{\Delta u}{\Delta t})$: the model uses only the discrete-time derivative $\frac{\Delta u}{\Delta t}$; (2) $f_v(u, \nabla u, \frac{\Delta u}{\Delta t})$: the model combines the discrete-time derivative with $u$ and $\nabla u$; (3) $f_v(u)$: the model uses only $u$; (4) $f_v(\nabla u)$: the model uses only $\nabla u$; (5) $f_v(u, \nabla u)$: the model relies solely on the wave function-derived $u$ and $\nabla u$. The results in Figure 3 demonstrate the effectiveness of the wave equation-informed approach. Specifically, (1) has an RMSE that is over 20% worse compared to (5). It is notable that experiment the incorporation of $\frac{\Delta u}{\Delta t}$ into the velocity model in (2) adversely affected performance compared to (5), primarily due to overfitting arising from the substantial discrepancy between the discrete-time derivative and the true values. Furthermore, the model in (5) outperforms (4), suggesting that the inclusion of $\nabla u$ with $u$ provides additional beneficial information to the network, enhancing its predictive capability. Figure 4 shows that WeatherODE produces much smoother $\frac{\Delta u}{\Delta t}$ predictions, aligning with the smooth nature of $u$, while the predictions of ClimODE are more erratic.

### 5.2 ANALYSIS OF SOURCE MODEL ARCHITECTURE

We conduct experiments by removing the source model and comparing different source model architectures: ViT, DiT, ResNet2D, and ResNet3D. DiT (Peebles & Xie, 2023) and ResNet3D are the time-aware versions of ViT and ResNet2D, respectively. As shown in Figure 5, DiT and ResNet3D outperform ViT and ResNet2D by 10% and 5%, and significantly exceed the performance of the

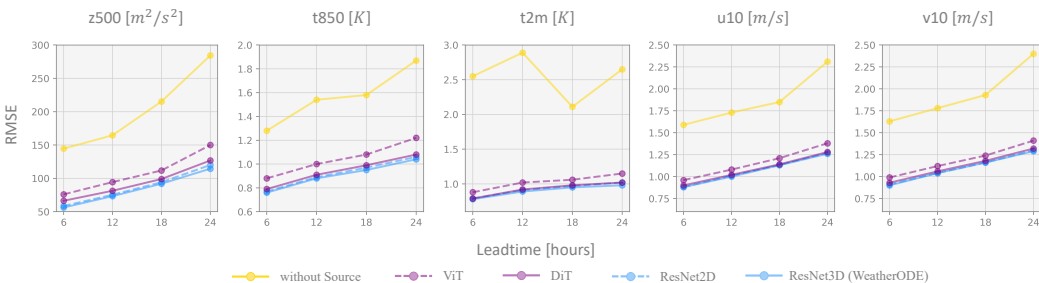

Figure 5: RMSE comparison for different architectures of the source model.

model without the source component. These results demonstrate the effectiveness of the source model and highlight the importance of integrating temporal information into its architecture.

## 5.3 STABILITY ANALYSIS OF NEURAL NETWORK AND NEURAL ODE INTEGRATION

The interdependencies between the advection and velocity models highlight the importance of carefully selecting architectures and learning rates to ensure the stability and performance of the neural network and neural ODE system. As shown in Table 3, the learning rate for the advection model must be lower than that of the velocity model due to often inaccurate initial estimates. If the advection model converges too quickly based on these estimates, it may lead to numerical instabilities and NaN values. Alternatively, using an advection model architecture with inherently slower convergence can yield similar results even with the same learning rate. Moreover, given that the source term represents solar energy with strong locality—where energy patterns are similar in neighboring regions—a CNN architecture that effectively captures local dependencies is ideal for this task.

| Velocity Model | Advection Model | Source Model | lr | Advection lr | Training Stable? | Rank |
|---|---|---|---|---|---|---|
| CNN | ViT | CNN | 5e-4 | 5e-4 | ✔ | 1 |
| ViT | ViT | CNN | 5e-4 | 5e-4 | ✔ | 4 |
| CNN | ViT | ViT | 5e-4 | 5e-4 | ✔ | 2 |
| ViT | ViT | ViT | 5e-4 | 5e-4 | ✔ | 5 |
| CNN | CNN | CNN | 5e-4 | 5e-4 | ✘ (1) | - |
| ViT | CNN | CNN | 5e-4 | 5e-4 | ✘ (1) | - |
| CNN | CNN | ViT | 5e-4 | 5e-4 | ✘ (1) | - |
| ViT | CNN | ViT | 5e-4 | 5e-4 | ✘ (1) | - |
| CNN | CNN | CNN | 5e-4 | 5e-5 | ✔ | 3 |
| ViT | CNN | ViT | 5e-4 | 5e-5 | ✘ (3) | - |
| ViT | CNN | ViT | 5e-4 | 5e-6 | ✔ | 6 |

Table 3: Stability analysis of neural network and neural ODE integration across different architectures and learning rates. "Advection lr" denotes the learning rate of the advection model and "lr" corresponds to the other two. "✔" indicates stable training, and "✘ $(i)$" shows where NaN values occurred at epoch $i$. "Rank" indicates the performance ranking among stable configurations.

## 6 CONCLUSION

In this paper, we tackle several challenges faced by neural ODE-based weather forecasting models, specifically addressing time-discretization errors, global-local biases across individual tasks in solving the advection equation, and discrepancies in time-dependent sources that compromise predictive accuracy. To address these issues, we present WeatherODE—a novel sandwich neural ODE model that integrates wave equation theory with a dynamic source model. This approach effectively reduces errors and promotes synergy between neural networks and neural ODEs. Our in-depth analysis of WeatherODE's architecture and optimization establishes a strong foundation for advancing hybrid modeling in meteorology. Looking forward, our work opens avenues for further exploration of hybrid models that blend traditional physics-driven and modern machine-learning techniques.

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

## A  ERA5 DATA

We train WeatherODE on the preprocessed $5.625°$ ERA5 data from WeatherBench (Rasp et al., 2020b), a benchmark dataset and evaluation framework designed to facilitate the comparison of data-driven weather forecasting models. WeatherBench regridded the raw ERA5 dataset[1] from its $0.25°$ resolution to three coarser resolutions: $5.625°$, $2.8125°$, and $1.40625°$. The processed dataset includes 8 atmospheric variables across 13 pressure levels, 6 surface variables, and 5 static variables. For training and testing WeatherODE, we selected 6 atmospheric variables at 7 pressure levels, 3 surface variables, and 3 static variables, as detailed in Table 4.

| Variable Name | Abbrev. | Description | Levels |
|---|---|---|---|
| Land-sea mask | $lsm$ | Binary mask distinguishing land (1) from sea (0) | N/A |
| Orography | $oro$ | Height of Earth's surface | N/A |
| Latitude | $lat$ | Latitude of each grid point | N/A |
| 2 metre temperature | $t2m$ | Temperature measured 2 meters above the surface | Single level |
| 10 metre U wind component | $u10$ | East-west wind speed at 10 meters above the surface | Single level |
| 10 metre V wind component | $v10$ | North-south wind speed at 10 meters above the surface | Single level |
| Geopotential | $z$ | Height relative to a pressure level | 50, 250, 500, 600, 700, 850, 925 hPa |
| U wind component | $u$ | Wind speed in the east-west direction | 50, 250, 500, 600, 700, 850, 925 hPa |
| V wind component | $v$ | Wind speed in the north-south direction | 50, 250, 500, 600, 700, 850, 925 hPa |
| Temperature | $t$ | Atmospheric temperature | 50, 250, 500, 600, 700, 850, 925 hPa |
| Specific humidity | $q$ | Mixing ratio of water vapor to total air mass | 50, 250, 500, 600, 700, 850, 925 hPa |
| Relative humidity | $r$ | Humidity relative to saturation | 50, 250, 500, 600, 700, 850, 925 hPa |

Table 4: Summary of ECMWF variables utilized in the ERA5 dataset. The variables $lsm$ and $oro$ are constant and invariant with time, while $t2m$, $u10$, and $v10$ represent surface variables. The remaining are atmospheric variables which are measured at specific pressure levels.

## B  WEATHER FORECASTING METRICS

In this section, we provide a detailed explanation of all the evaluation metrics used in Section 4. For each metric, $u$ and $\tilde{u}$ represent the predicted and ground truth values, respectively, both shaped as $K \times H \times W$, where $K$ is the number of predict quantities, and $H \times W$ is the spatial resolution. The latitude weighting term $\alpha(\cdot)$ accounts for the non-uniform grid cell areas.

**Latitude-weighted Root Mean Square Error (RMSE)**  assesses model accuracy while considering the Earth's curvature. The latitude weighting adjusts for the varying grid cell areas at different latitudes, ensuring that errors are appropriately measured. Lower RMSE values indicate better model performance.

$$\text{RMSE} = \frac{1}{K} \sum_{k=1}^{K} \sqrt{\frac{1}{HW} \sum_{h=1}^{H} \sum_{w=1}^{W} \alpha(h) \left(\tilde{u}_{k,h,w} - u_{k,h,w}\right)^2}, \ \alpha(h) = \frac{\cos(\text{lat}(h))}{\frac{1}{H} \sum_{h'=1}^{H} \cos\left(\text{lat}\left(h'\right)\right)}.$$

**Anomaly Correlation Coefficient (ACC)**  measures a model's ability to predict deviations from the mean. Higher ACC values indicate better accuracy in capturing anomalies, which is crucial in meteorology and climate science.

$$\text{ACC} = \frac{\sum_{k,h,w} \tilde{u}'_{k,h,w} u'_{k,h,w}}{\sqrt{\sum_{k,h,w} \alpha(h)(\tilde{u}'_{k,h,w})^2 \sum_{k,h,w} \alpha(h)(u'_{k,h,w})^2}},$$

where $u' = u - C$ and $\tilde{u}' = \tilde{u} - C$, with $C = \frac{1}{K} \sum_k \tilde{u}_k$ representing the temporal mean of the ground truth over the test set.

---

[1] For more details of the raw ERA5 data, see `https://confluence.ecmwf.int/display/CKB/ERA5%3A+data+documentation`.

## C IMPLEMENTATION DETAILS

### C.1 DATA FLOW

We normalized all inputs by computing the mean and standard deviation for each variable at each pressure level (for atmospheric variables) to achieve zero mean and unit variance. After normalization, the input $u(t_0) \in \mathbb{R}^{K \times H \times W}$ with its spatial derivative $\nabla u(t_0) \in \mathbb{R}^{2K \times H \times W}$ are processed by the velocity model $f_v(\cdot)$ to estimate the initial velocity $v_0 \in \mathbb{R}^{2K \times H \times W}$. Both $u(t_0)$ and $v(t_0)$ are then fed into the ODE system, where the $\dot{u}(t_n)$ is calculated by advection equation and the advection model $f_\theta(\cdot)$ uses $u(t_n), \nabla u(t_n), v(t_n)$ and $(\phi_s, \phi_t)$ to model $\dot{v}(t_n)$. The ODE system outputs the predicted future state $\{u(t_n)\}_{n=1}^N$, where $N$ represents the lead time.

The predicted $\{u(t_n)\}_{n=1}^N$, along with $u(t_0)$, and $v(t_0)$, are then passed into the source model $f_s(\cdot)$ to estimate the source term $\{s(t_n)\}_{n=1}^N$. The final prediction for the lead time and each intermediate time point is obtained by adding $s(t_n)$ to $u(t_n)$ and then applying the inverse normalization. For training and evaluation, we selected five key variables from the $K$ input variables: $z500, t850, t2m, u10,$ and $v10$.

### C.2 EMBEDDINGS

**Spatial Encoding** Latitude $h$ and longitude $w$ are encoded with trigonometric and spherical coordinates:

$$\phi_s = [\sin(h), \cos(h), \sin(w), \cos(w), \sin(h)\cos(w), \sin(h)\sin(w)].$$

**Temporal Encoding** Daily and seasonal cycles are encoded using trigonometric functions:

$$\phi_t = \left[\sin(2\pi t), \cos(2\pi t), \sin\left(\frac{2\pi t}{365}\right), \cos\left(\frac{2\pi t}{365}\right)\right].$$

**Spatiotemporal Embedding** The final embedding integrates both:

$$(\phi_s, \phi_t) = [\phi_s, \phi_t, \phi_s \times \phi_t].$$

### C.3 OPTIMIZATION

All experiments are conducted with a batch size of 8, running on 4 NVIDIA A800-SXM4-80GB GPUs for 50 epochs. We use the AdamW optimizer with $\beta_1 = 0.9$, $\beta_2 = 0.999$. The learning rate is set to $1e\text{-}4$ for the ODE model components and $5e\text{-}4$ for the rest. A weight decay of $1e\text{-}5$ is applied to all parameters except for the positional embeddings. The learning rate follows a linear warmup schedule starting from $1e\text{-}8$ for the first $10,000$ steps (approximately 1 epoch), transitioning to a cosine-annealing schedule for the remaining $90,000$ steps (approximately 9 epochs), with a minimum value of $1e\text{-}8$.

### C.4 HYPERPARAMETERS

| Hyperparameter | Description | Value |
|---|---|---|
| Kernel size | Size of each convolutional kernels | 3 |
| Padding size | Size of padding of each convolutional layer | 1 |
| Stride | Step size of each convolutional layer | 1 |
| Dropout | Dropout probability | 0.1 |
| Leakage Coefficient | Slope of LeakyReLU for negative inputs | 0.3 |
| ResBlock List | (Number of ResBlocks, Hidden dimensions) | $[(5, 512), (5, 128), (3, 64), (2, 48)]$ |

Table 5: Default hyperparameters of ResNet2D of velocity model.

| Hyperparameter | Description | Value |
|---|---|---|
| $p$ | Size of image patches | 2 |
| $D$ | Dimension of hidden layers | 1024 |
| Depth | Number of Transformer blocks | 4 |
| Heads | Number of attention heads | 8 |
| MLP ratio | Expansion factor for MLP | 4 |
| Decoder Depth | Number of layers of the final prediction head | 2 |
| Drop path | Stochastic depth rate | 0.1 |
| Dropout | Dropout rate | 0.1 |

Table 6: Default hyperparameters of ViT in advection ODE.

| Hyperparameter | Description | Value |
|---|---|---|
| Kernel size | Size of each 3D convolutional kernels | 3 |
| Padding size | Size of padding of each 3D convolutional layer | 1 |
| Stride | Step size of each 3D convolutional layer | 1 |
| Dropout | Dropout probability | 0.1 |
| Leakage Coefficient | Slope of LeakyReLU for negative inputs | 0.3 |
| ResBlock List | (Number of ResBlocks, Hidden dimensions) | $[(5, 512), (5, 128), (3, 64), (2, 48)]$ |

Table 7: Default hyperparameters of ResNet3D of source model.

## D   REGIONAL FORECAST

Obtaining global data is often challenging, making it crucial to develop methods that can predict weather using data from specific local regions. As shown in Figure 6, we illustrate the forecasting pipeline for regional prediction. We conduct experiments focusing on three regions: North America, South America, and Australia. The data for these regions is extracted as bounding boxes from the $5.625°$ ERA5 global dataset. Table 8 provides the bounding box details for each of the three regions.

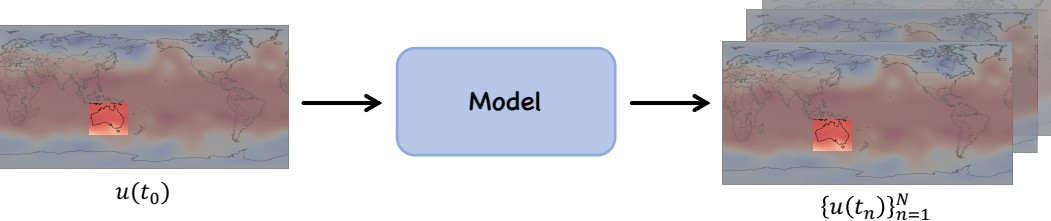

$u(t_0)$                  $\{u(t_n)\}_{n=1}^N$

Figure 6: Schematic of the regional forecast for Australia, where only data from the Australian region is used to predict weather conditions within the same area.

| Region | Latitude Range | Longitude Range | Grid Size (lat x lon) |
|---|---|---|---|
| North America | $(15, 65)$ | $(220, 300)$ | $8 \times 14$ |
| South America | $(-55, 20)$ | $(270, 330)$ | $14 \times 10$ |
| Australia | $(-50, 10)$ | $(100, 180)$ | $10 \times 14$ |
| Global | $(-90, 90)$ | $(0, 360)$ | $32 \times 64$ |

Table 8: Latitudinal and longitudinal boundaries with grid size for each region.

# E    FULL RESULTS

| Variable | Hours | RMSE ↓ | | | ACC ↑ | | |
|---|---|---|---|---|---|---|---|
| | | ClimaX | ClimODE | WeatherODE | ClimaX | ClimODE | WeatherODE |
| z500 | 36 | 126.4 | 259.6 | 159.9 | 1.00 | 0.96 | 0.99 |
| | 72 | 244.1 | 478.7 | 324.7 | 1.00 | 0.88 | 0.95 |
| t850 | 36 | 1.25 | 1.75 | 1.22 | 0.97 | 0.94 | 0.97 |
| | 72 | 1.59 | 2.58 | 1.81 | 0.98 | 0.85 | 0.93 |
| t2m | 36 | 1.33 | 1.70 | 1.18 | 0.97 | 0.94 | 0.97 |
| | 72 | 1.43 | 2.75 | 1.60 | 0.98 | 0.85 | 0.95 |
| u10 | 36 | 1.57 | 2.25 | 1.57 | 0.93 | 0.83 | 0.93 |
| | 72 | 2.18 | 3.19 | 2.45 | 0.94 | 0.66 | 0.81 |
| v10 | 36 | (N/A) | 2.29 | 1.61 | (N/A) | 0.83 | 0.92 |
| | 72 | (N/A) | 3.30 | 2.50 | (N/A) | 0.63 | 0.80 |

Table 9: Latitude-weighted RMSE and ACC for global forecasting at longer lead times.

| Variable | Hours | RMSE ↓ | | | | | ACC ↑ | | | | |
|---|---|---|---|---|---|---|---|---|---|---|---|
| | | wo Source | ViT | DiT | Resnet2D | Resnet3D | wo Source | ViT | DiT | ResNet2D | Resnet3D |
| z500 | 6 | 144.6 | 76.0 | 66.4 | 58.6 | 56.3 | 0.99 | 1.00 | 1.00 | 1.00 | 1.00 |
| | 12 | 164.6 | 94.4 | 81.4 | 75.4 | 73.3 | 0.99 | 1.00 | 1.00 | 1.00 | 1.00 |
| | 18 | 215.3 | 111.8 | 99.0 | 94.2 | 91.9 | 0.98 | 0.99 | 1.00 | 1.00 | 1.00 |
| | 24 | 284.3 | 150.0 | 126.8 | 120.0 | 114.5 | 0.96 | 0.99 | 0.99 | 0.99 | 1.00 |
| t850 | 6 | 1.28 | 0.88 | 0.79 | 0.77 | 0.76 | 0.97 | 0.98 | 0.99 | 0.99 | 0.99 |
| | 12 | 1.54 | 1.00 | 0.91 | 0.89 | 0.88 | 0.95 | 0.98 | 0.98 | 0.98 | 0.98 |
| | 18 | 1.58 | 1.08 | 0.99 | 0.97 | 0.95 | 0.95 | 0.98 | 0.98 | 0.98 | 0.98 |
| | 24 | 1.87 | 1.22 | 1.08 | 1.06 | 1.04 | 0.93 | 0.97 | 0.98 | 0.98 | 0.98 |
| t2m | 6 | 2.55 | 0.88 | 0.79 | 0.79 | 0.78 | 0.87 | 0.99 | 0.99 | 0.99 | 0.99 |
| | 12 | 2.89 | 1.02 | 0.92 | 0.91 | 0.89 | 0.83 | 0.98 | 0.98 | 0.98 | 0.98 |
| | 18 | 2.11 | 1.06 | 0.98 | 0.97 | 0.95 | 0.91 | 0.98 | 0.98 | 0.98 | 0.98 |
| | 24 | 2.65 | 1.15 | 1.02 | 1.02 | 0.98 | 0.86 | 0.97 | 0.98 | 0.98 | 0.98 |
| u10 | 6 | 1.59 | 0.96 | 0.90 | 0.88 | 0.88 | 0.92 | 0.97 | 0.98 | 0.98 | 0.98 |
| | 12 | 1.73 | 1.08 | 1.02 | 1.01 | 1.00 | 0.91 | 0.96 | 0.97 | 0.97 | 0.97 |
| | 18 | 1.85 | 1.21 | 1.14 | 1.13 | 1.13 | 0.89 | 0.95 | 0.96 | 0.96 | 0.96 |
| | 24 | 2.31 | 1.38 | 1.28 | 1.27 | 1.26 | 0.84 | 0.94 | 0.95 | 0.95 | 0.95 |
| v10 | 6 | 1.63 | 0.99 | 0.93 | 0.91 | 0.90 | 0.92 | 0.97 | 0.97 | 0.98 | 0.98 |
| | 12 | 1.78 | 1.12 | 1.06 | 1.04 | 1.04 | 0.90 | 0.97 | 0.97 | 0.97 | 0.97 |
| | 18 | 1.93 | 1.24 | 1.18 | 1.16 | 1.16 | 0.89 | 0.96 | 0.96 | 0.96 | 0.96 |
| | 24 | 2.40 | 1.41 | 1.32 | 1.30 | 1.29 | 0.82 | 0.94 | 0.95 | 0.95 | 0.95 |

Table 10: Full results on source model architectures shown in Figure 5.

| | | RMSE ↓ | | | | | ACC ↑ | | | | |
|---|---|---|---|---|---|---|---|---|---|---|---|
| Variable | Hours | $f_v(\frac{\Delta u}{\Delta t})$ | $f_v(u)$ | $f_v(\nabla u)$ | $f_v(u,\nabla u,\frac{\Delta u}{\Delta t})$ | $f_v(u,\nabla u)$ | $f_v(\frac{\Delta u}{\Delta t})$ | $f_v(u)$ | $f_v(\nabla u)$ | $f_v(u,\nabla u,\frac{\Delta u}{\Delta t})$ | $f_v(u,\nabla u)$ |
| $z500$ | 6 | 71.0 | 73.6 | 59.8 | 61.4 | 56.3 | 1.00 | 1.00 | 1.00 | 1.00 | 1.00 |
| | 12 | 100.6 | 101.2 | 79.4 | 83.5 | 73.3 | 0.99 | 1.00 | 1.00 | 1.00 | 1.00 |
| | 18 | 134.0 | 129.7 | 101.6 | 108.1 | 91.9 | 0.99 | 0.99 | 1.00 | 0.99 | 1.00 |
| | 24 | 172.8 | 162.6 | 128.5 | 137.6 | 114.5 | 0.98 | 0.99 | 0.99 | 0.99 | 1.00 |
| $t850$ | 6 | 0.83 | 0.83 | 0.77 | 0.79 | 0.76 | 0.99 | 0.99 | 0.99 | 0.99 | 0.99 |
| | 12 | 1.00 | 0.98 | 0.90 | 0.92 | 0.88 | 0.98 | 0.98 | 0.98 | 0.98 | 0.98 |
| | 18 | 1.12 | 1.09 | 0.99 | 1.02 | 0.95 | 0.97 | 0.98 | 0.98 | 0.98 | 0.98 |
| | 24 | 1.25 | 1.20 | 1.08 | 1.12 | 1.04 | 0.97 | 0.97 | 0.98 | 0.97 | 0.98 |
| $t2m$ | 6 | 0.91 | 0.89 | 0.80 | 0.82 | 0.78 | 0.98 | 0.98 | 0.99 | 0.99 | 0.99 |
| | 12 | 1.09 | 1.04 | 0.93 | 0.95 | 0.89 | 0.98 | 0.98 | 0.98 | 0.98 | 0.98 |
| | 18 | 1.13 | 1.10 | 0.99 | 1.02 | 0.95 | 0.98 | 0.98 | 0.98 | 0.98 | 0.98 |
| | 24 | 1.16 | 1.13 | 1.03 | 1.06 | 0.98 | 0.97 | 0.98 | 0.98 | 0.98 | 0.98 |
| $u10$ | 6 | 0.96 | 0.95 | 0.89 | 0.90 | 0.88 | 0.97 | 0.97 | 0.98 | 0.98 | 0.98 |
| | 12 | 1.13 | 1.12 | 1.03 | 1.04 | 1.00 | 0.96 | 0.96 | 0.97 | 0.97 | 0.97 |
| | 18 | 1.30 | 1.28 | 1.15 | 1.18 | 1.13 | 0.95 | 0.95 | 0.96 | 0.96 | 0.96 |
| | 24 | 1.50 | 1.45 | 1.30 | 1.33 | 1.26 | 0.93 | 0.94 | 0.95 | 0.95 | 0.95 |
| $v10$ | 6 | 0.98 | 0.98 | 0.92 | 0.93 | 0.90 | 0.97 | 0.97 | 0.97 | 0.97 | 0.98 |
| | 12 | 1.16 | 1.16 | 1.06 | 1.07 | 1.04 | 0.96 | 0.96 | 0.97 | 0.97 | 0.97 |
| | 18 | 1.34 | 1.32 | 1.19 | 1.21 | 1.16 | 0.95 | 0.95 | 0.96 | 0.96 | 0.96 |
| | 24 | 1.54 | 1.50 | 1.33 | 1.37 | 1.29 | 0.93 | 0.93 | 0.95 | 0.94 | 0.95 |

Table 11: Full results of different input configurations of the velocity model in Figure 3.

| | | RMSE ↓ | | | | | ACC ↑ | | | | |
|---|---|---|---|---|---|---|---|---|---|---|---|
| Variable | Hours | $\Delta t = 1$ | $\Delta t = 2$ | $\Delta t = 3$ | $\Delta t = 12$ | $\Delta t = 24$ | $\Delta t = 1$ | $\Delta t = 2$ | $\Delta t = 3$ | $\Delta t = 12$ | $\Delta t = 24$ |
| $z500$ | 6 | 71.0 | 86.8 | 88.8 | 107.8 | 140.5 | 1.00 | 1.00 | 1.00 | 0.99 | 0.98 |
| | 12 | 100.6 | 118.5 | 128.7 | 164.3 | NaN | 0.99 | 0.99 | 0.99 | 0.98 | NaN |
| | 18 | 134.0 | 157.3 | 161.3 | NaN | NaN | 0.99 | 0.99 | 0.99 | NaN | NaN |
| | 24 | 172.8 | NaN | NaN | NaN | NaN | 0.98 | NaN | NaN | NaN | NaN |
| $t850$ | 6 | 0.83 | 0.89 | 0.91 | 0.95 | 1.04 | 0.99 | 0.98 | 0.98 | 0.97 | 0.97 |
| | 12 | 1.00 | 1.10 | 1.11 | 1.18 | NaN | 0.98 | 0.98 | 0.97 | 0.97 | NaN |
| | 18 | 1.12 | 1.24 | 1.25 | NaN | NaN | 0.97 | 0.97 | 0.97 | NaN | NaN |
| | 24 | 1.25 | NaN | NaN | NaN | NaN | 0.97 | NaN | NaN | NaN | NaN |
| $t2m$ | 6 | 0.91 | 0.98 | 1.00 | 1.12 | 1.35 | 0.98 | 0.98 | 0.98 | 0.97 | 0.96 |
| | 12 | 1.09 | 1.18 | 1.23 | 1.35 | NaN | 0.98 | 0.97 | 0.97 | 0.96 | NaN |
| | 18 | 1.13 | 1.21 | 1.28 | NaN | NaN | 0.98 | 0.97 | 0.97 | NaN | NaN |
| | 24 | 1.16 | NaN | NaN | NaN | NaN | 0.97 | NaN | NaN | NaN | NaN |
| $u10$ | 6 | 0.96 | 1.02 | 1.03 | 1.07 | 1.11 | 0.97 | 0.97 | 0.97 | 0.97 | 0.96 |
| | 12 | 1.13 | 1.24 | 1.23 | 1.41 | NaN | 0.96 | 0.95 | 0.95 | 0.93 | NaN |
| | 18 | 1.30 | 1.44 | 1.42 | NaN | NaN | 0.95 | 0.93 | 0.94 | NaN | NaN |
| | 24 | 1.50 | NaN | NaN | NaN | NaN | 0.93 | NaN | NaN | NaN | NaN |
| $v10$ | 6 | 0.98 | 1.06 | 1.07 | 1.11 | 1.16 | 0.97 | 0.97 | 0.96 | 0.96 | 0.96 |
| | 12 | 1.16 | 1.28 | 1.31 | 1.44 | NaN | 0.96 | 0.95 | 0.94 | 0.94 | NaN |
| | 18 | 1.34 | 1.47 | 1.52 | NaN | NaN | 0.95 | 0.93 | 0.93 | NaN | NaN |
| | 24 | 1.54 | NaN | NaN | NaN | NaN | 0.93 | NaN | NaN | NaN | NaN |

Table 12: Full result of the time interval $\Delta t$ for estimating $\frac{\Delta u}{\Delta t}$ in Figure 1b. NaN indicates that numerical instability occurred during ODE inference.

# F VISUALIZATION

Figure 7 to Figure 10 provide visual comparisons between WeatherODE's forecasts and the ground truth ERA5 data at different lead times (6h, 12h, 18h, and 24h). Figure 11 illustrates the output from the advection ODE and the source model, demonstrating that the advection ODE captures global features, while the source model captures local features.

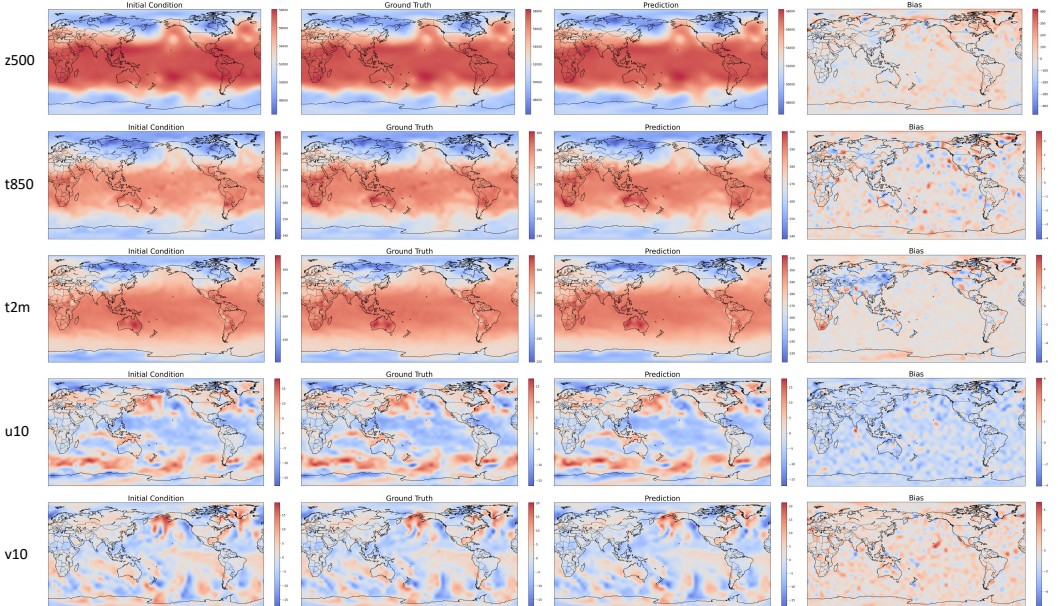

Figure 7: Example 6-hour lead time forecasts from WeatherODE compared to ground truth ERA5 data.

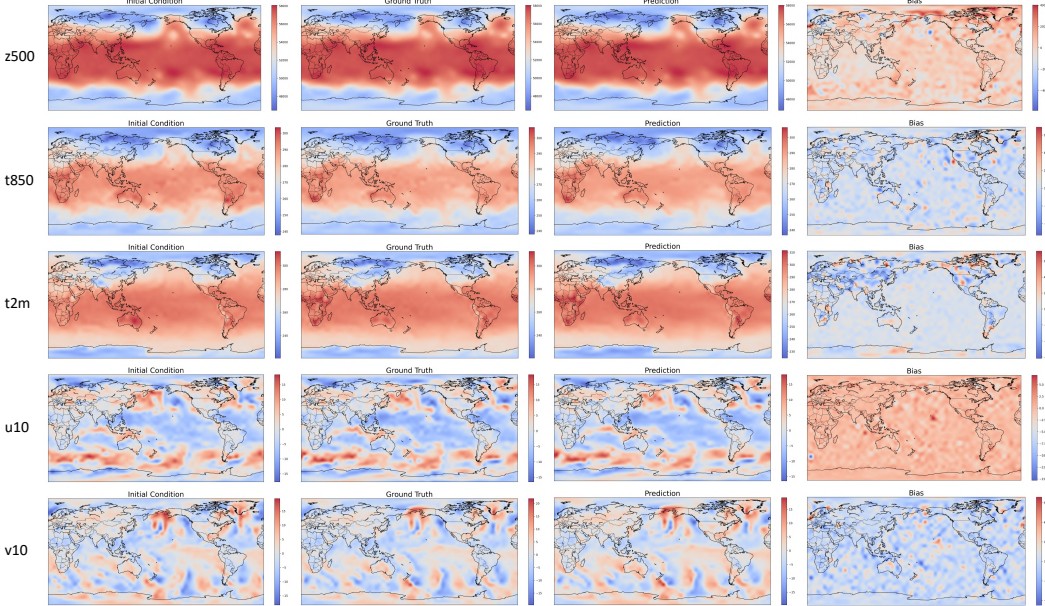

Figure 8: Example 12-hour lead time forecasts from WeatherODE compared to ground truth ERA5 data.

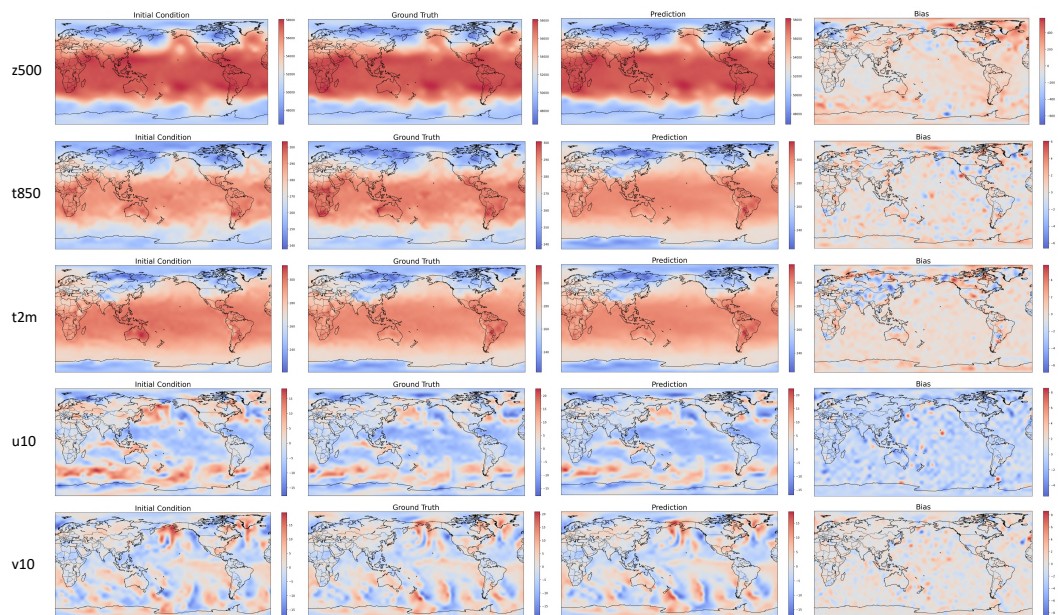

Figure 9: Example 18-hour lead time forecasts from WeatherODE compared to ground truth ERA5 data.

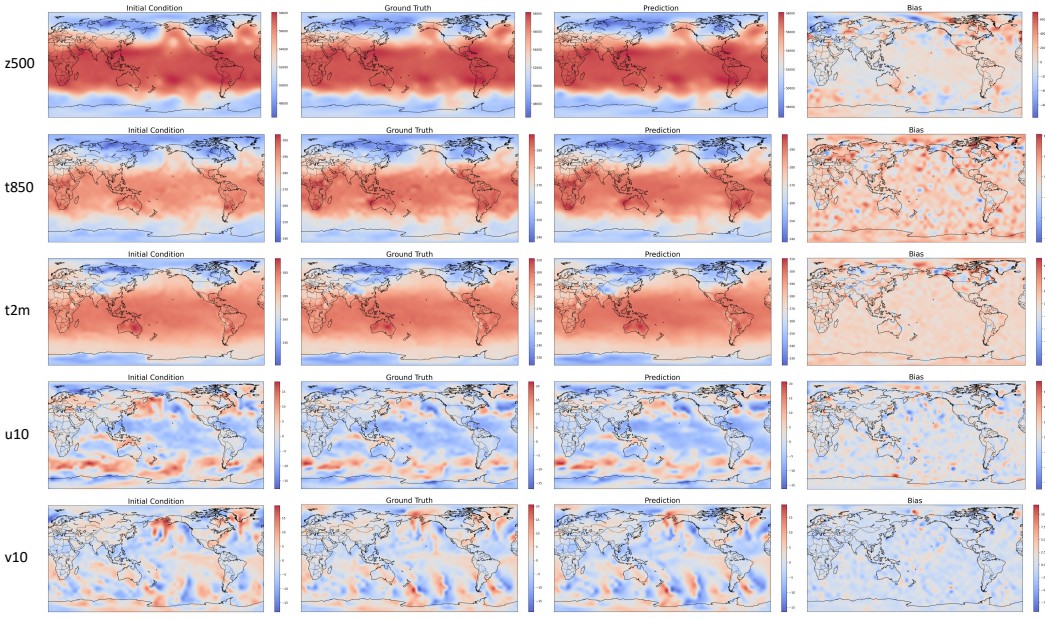

Figure 10: Example 24-hour lead time forecasts from WeatherODE compared to ground truth ERA5 data.

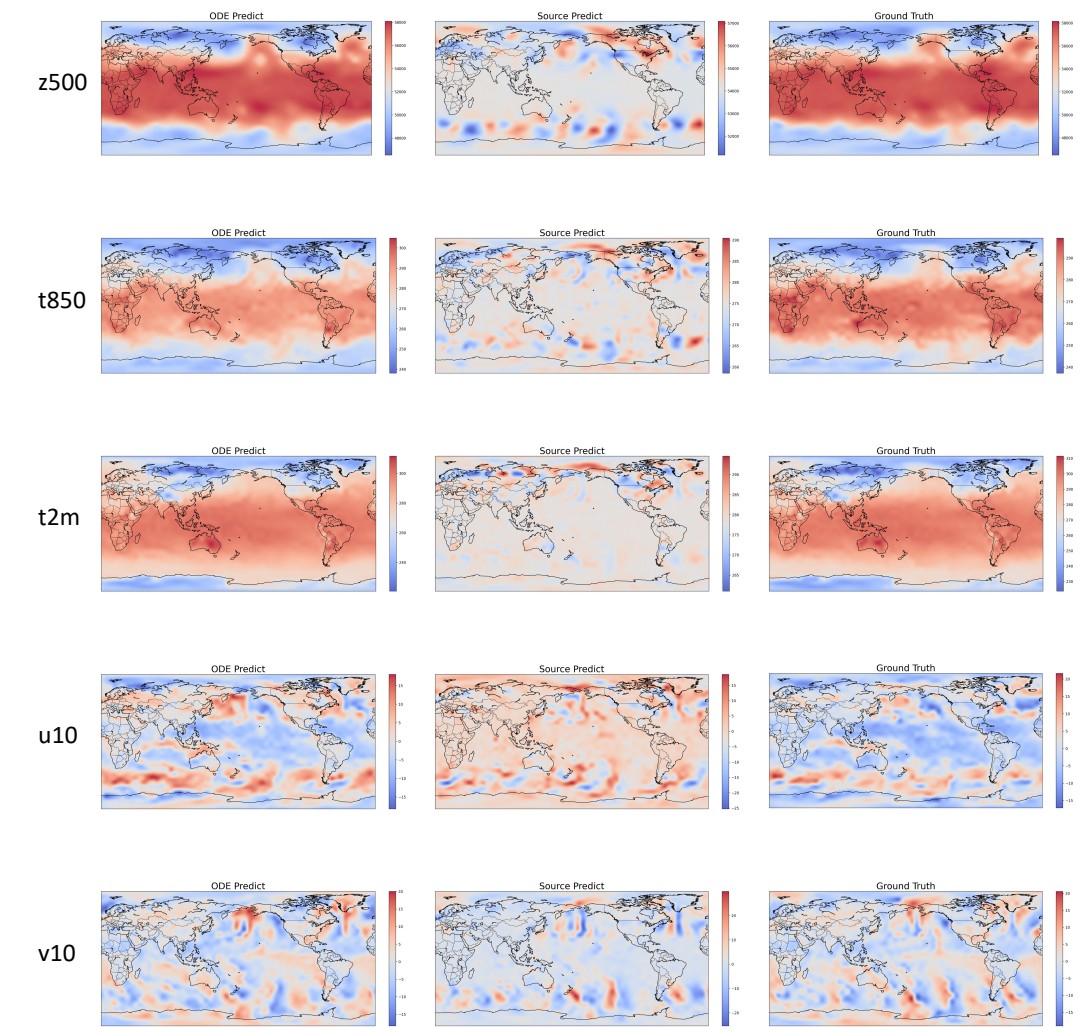

Figure 11: Example 24-hour lead time forecasts from the advection ODE, source model, and ground truth comparison.

