# OpenReview forum: "Mitigating Time Discretization Challenges with WeatherODE: A Sandwich Physics-Driven Neural ODE for Weather Forecasting"
_ICLR.cc/2025/Conference — Submitted to ICLR 2025_

### Official Review · Reviewer_o4jc · 2024-10-17

**Soundness:** 3
**Presentation:** 2
**Contribution:** 3
**Rating:** 3
**Confidence:** 4

**Summary:**

The paper extends a recent neural weather ODE model (ClimODE 2024) by using wave-equations for initial state estimation, adopting a more transformer based dynamics, and adding a CNN source field. The contributions range from incremental (transformers) to substantial (source field). The results are outstanding.

------

Post-response update. I'm decreasing the score to reject, which I understand is exceptional given my initial positive review. The authors have not been able to clarify or address my concerns regarding the role of the source term, and the role of wave term in the advective system. I believe using the wave term is unfounded in an advective system, while the source term seems not to be part of the PDE afterall (given that source gets as input future states), which violates eq 1. The paper is not ready for publication.

**Strengths:**

- The paper extends weather ODEs in sensible ways, and the overall model is sensible. Adding the source field is a major contribution, while the initial estimation and network searches seem useful (but more incremental).
- The results are outstanding: they even beat IFS at times, which is an outstanding achievement!

**Weaknesses:**

- The motivations behind the model choices seem a bit weak
- The ablations are very interesting, but have some issues. In the v0 ablation there is no ClimODE baseline, so it’s difficult to say if any improvement was actually gained here. The Fig4 is superficial and difficult to interpret. The stability analysis seems interesting, but I don’t think it provides much insights into why things sometimes fail. One would not really expect such a simple ODE system to fail in the first place.
- The experiments make an unfair comparison to the ClimODE baseline, where ClimODE uses 5 data variables and weatherODE 48. Results lack standard deviations. Some results show that weatherODE beats IFS, and this is not elaborated further. This is a major achievement, and is now provided a bit too casually. There should be some discussion on how the results relate to the larger model (panguweather, gencast, graphcast, etc).
- The clarity and text needs improvements.

**Questions:**

- I have hard time understanding the v0 estimation. First, a wave equation is introduced in general terms, and then a CNN appears. These two are not connected together, so I’m left wondering what do we do with the waves, and how do they relate to the CNN. Furthermore, the wave equation is poorly motivated, and it seems disconnected from the advection equation 3. To me this looks like a discrepancy: the system follows advection, but initial state assumes different kind of physics (ie. laplacians appear out of nowhere). Surely the initial state estimation needs to respect the chosen ODE model, and not introduce some physics effects that are not part of the ODE.
- The arguments about spatial vs temporal resolution are not convincing. Low temporal sampling rate does not necessarily mean that the temporal resolution is low: this depends on how quickly the process varies. It also feels misguided to say that spatial resolution is 100x higher than temporal resolution: this is only true if you take 32*64, and I don’t think you should do this. Since the (t,x,y) axes have similar ranges (24,32,64), I would argue that there is no resolution gap in the sampling rate between space and time.
- I couldn’t follow the CNN/ViT convergence arguments. I’m not sure what convergence even means here (of training..?, of rollouts..?). I think the paper is arguing that training CNNs is somehow instable, and thus ViT’s have to be used. This sounds implausible, and a poor motivation for choosing how the dynamics should evolve. Surely the networks need to be chosen such that they respect some physical system properties (eg. feature locality/globality).
- I don’t understand the source model. It takes as input all 1…N simulated states u(t_n), but one needs the source states s(t_n) to do this simulation. This has to be a mistake: I assume that we instead take as inputs u(t_1 : t_n), so that the history grows along the ODE rollouts. How do you handle set inputs and set outputs? I don’t think a 3D CNN supports sets as inputs or outputs [maybe these are not sets, but just tensors of fixed size..].
- At sec 3.5. I’m again confused what do you mean by “convergence”.
- Sec 3.5. claims that advection dynamics involves long-range dependencies. I’m not sure I agree, and would even argue the opposite. PDE’s are by nature local models: why would the weather in new york affect the infinitesimal change of weather in london? There is no connection between them. Can you elaborate?
- Sec 3.6. claims that earlier methods only train against final state, and ignore intermediate terms. I’m surprised by this statement. ClimODE trains with all intermediate points, as does the original neural ODE, and every other neural ODE/PDE model I’ve seen. Can you provide examples of models where this happens? Calling this “multi-task” learning is also wrong: intermediate points in an ODE are not different “tasks” (they are not even a single repeated task, since it’s not a “task” in the first place).
- The benchmarks use 48 weather variables, but only evaluate 5 of them. The benchmarks also take ClimODE results as-is from the paper at least in Table 2. The results are then unfair to ClimODE: the ClimODE is using only 5 variables worth of information, while weatherODE is using 48 variables. The paper needs compare apple to apples by either running weatherODE results using only 5 variables, or running ClimODE with 48 variables.

I'm looking forward to the responses.

---

> ### Author Response · Authors · 2024-11-18
> **Response**
>
> We greatly appreciate Reviewer o4jc's thoughtful feedback and their recognition of the meaningful extensions we’ve made to ODEs for weather forcasting. Below, we address each point raised, offering clarifications and detailed responses.
>
> # **Weakness**
>
> > The motivations behind the model choices seem a bit weak.
>
> The motivation behind our model choices stems from the core idea of the Physical-informed Neural ODE framework, particularly the advection function proposed in ClimODE. To solve this ODE, we need to accurately estimate the initial velocity, solve the advection equation, and finally account for the error term arising from deviations in real-world conditions. These three tasks are inherently different due to their distinct physical natures. **Thus, we aim to design an end-to-end architecture that can model this entire process seamlessly. WeatherODE, as proposed, provides an integrated end-to-end solution that handles all three tasks, while ensuring numerical stability during training.**
>
> > The ablations are very interesting, but have some issues. In the v0 ablation there is no ClimODE baseline, so it’s difficult to say if any improvement was actually gained here. The Fig4 is superficial and difficult to interpret. The stability analysis seems interesting, but I don’t think it provides much insights into why things sometimes fail. One would not really expect such a simple ODE system to fail in the first place.
>
> - Regarding the v0 ablation, we add the missing results in **Table A** below. In this ablation, we modify only the velocity network in ClimODE to handle different inputs and used the full ERA5 dataset for training. Everything else remains identical to ClimODE. **The results are consistent with those in Figure 3 of our paper,** showing that denser spatial information helps reduce discretization errors in velocity prediction.
> - As for Figure 4, we aim to visualize the evolution of the weather variable $u$ from the ERA5 dataset over time, alongside the estimated velocity $v$ using ClimODE (approximated by $\frac{\Delta u}{\Delta t}$) and WeatherODE (derived from wave equation theory as $\frac{\partial u}{\partial t}$). **The purpose is to highlight the continuous nature of $u$ over time, with WeatherODE providing a much smoother and more consistent derivative compared to ClimODE.**
> - Apologies for the lack of clarity in the stability analysis; it doesn’t fully explain why the system fails in some cases. **The primary issue is sensitivity to initial conditions. If the initial velocity $v_0$ is poorly estimated, errors accumulate quickly, leading to blow-up (NaN values), where the values become either excessively large or small, causing numerical instability.** ClimODE mitigates this by pre-training the velocity network for better $v_0$ estimation and reducing ODE solver precision. WeatherODE, on the other hand, is an end-to-end training framework, where the velocity network must converge faster than the neural ODE to avoid instability, as discussed in Section 5.3. This ensures the network provides accurate initial velocity estimates from the start, preventing numerical blow-up.
>
> **Table A.** Experiment of estimation of $v_0$ of ClimODE
> |||RMSE|||ACC|||
> |-|-|-|-|-|-|-|-|
> |Variable|Lead Time|$f_v(\frac{\Delta u}{\Delta t})$ (ClimODE)|$f_v(\Delta u)$|$f_v(u, \Delta u)$|$f_v(\frac{\Delta u}{\Delta t})$ (ClimODE)|$f_v(\Delta u)$|$f_v(u, \Delta u)$|
> |z500|6|79.1|74.2|68.7|1.00|1.00|1.00|
> ||12|111.2|97.6|88.5|0.99|1.00|1.00|
> ||18|148.9|125.1|110.5|0.99|0.99|1.00|
> ||24|192.8|158.0|138.8|0.98|0.99|0.99|
> |t850|6|1.01|0.96|0.92|0.98|0.99|0.99|
> ||12|1.23|1.09|1.09|0.97|0.98|0.98|
> ||18|1.40|1.19|1.14|0.96|0.98|0.98|
> ||24|1.52|1.34|1.28|0.95|0.97|0.98|
> |t2m|6|1.13|1.00|0.97|0.97|0.98|0.99|
> ||12|1.36|1.13|1.07|0.97|0.98|0.98|
> ||18|1.54|1.22|1.19|0.96|0.98|0.98|
> ||24|1.75|1.28|1.18|0.96|0.98|0.98|
> |u10|6|1.15|1.09|1.10|0.95|0.97|0.98|
> ||12|1.41|1.26|1.23|0.94|0.96|0.97|
> ||18|1.68|1.41|1.37|0.93|0.95|0.96|
> ||24|1.83|1.61|1.53|0.90|0.94|0.95|
> |v10|6|1.21|1.12|1.08|0.95|0.97|0.97|
> ||12|1.47|1.30|1.29|0.94|0.96|0.97|
> ||18|1.65|1.47|1.39|0.93|0.95|0.96|
> ||24|1.86|1.62|1.55|0.90|0.93|0.95|

---

> ### Author Response · Authors · 2024-11-18
> **Response Continued**
>
> > The experiments make an unfair comparison to the ClimODE baseline, where ClimODE uses 5 data variables and weatherODE 48. Results lack standard deviations. Some results show that weatherODE beats IFS, and this is not elaborated further. This is a major achievement, and is now provided a bit too casually. There should be some discussion on how the results relate to the larger model (panguweather, gencast, graphcast, etc).
>
> In response, we add experiments where WeatherODE uses only 5 input variables in **Table B** below. The results continue to outperform ClimODE. The standard deviation in ClimODE's results primarily arises from the use of a source network that outputs a standard deviation term as a loss function regularization, which we have omitted in our design.
>
> Regarding WeatherODE beating IFS, this was observed for the $t2m$ variable. For other variables, the performance is slightly below that of IFS. It's important to note that IFS results were trained on data at a 0.25° resolution, while WeatherODE was trained at 5.625° resolution, yet it still provides comparable 24-hour forecasts. **This demonstrates the potential of Neural ODEs in weather prediction, even at coarser resolutions.**
>
> As for models like Pangu, GenCast, and GraphCast, they are developed by large organizations (e.g., Google, Huawei) with access to hundreds or thousands of GPUs, and trained on 0.25° data, **which presents a resource challenge for us.** However, **it's worth noting that scaling up has been widely validated in fields like CV and NLP.** We believe that WeatherODE, once scaled up, can achieve results comparable to these models even at 0.25° resolution.
>
> **Table B.** Experiment of estimation of WeatherODE with only 5 variables
>
> |||RMSE|||ACC|||
> |-|-|-|-|-|-|-|-|
> |Variable|Lead Time|ClimODE|WeatherODE* (5 variable)|ClimODE|WeatherODE* (5 variable)|
> |z500|6|102.9|71.5|0.99|1.00|
> ||12|134.8|102.9|0.99|0.99|
> ||18|162.7|135.9|0.98|0.99|
> ||24|193.4|171.8|0.98|0.99|
> |t850|6|1.16|0.84|0.97|0.99|
> ||12|1.32|1.01|0.96|0.98|
> ||18|1.47|1.11|0.96|0.97|
> ||24|1.55|1.21|0.95|0.97|
> |t2m|6|1.21|0.86|0.97|0.99|
> ||12|1.45|1.01|0.96|0.98|
> ||18|1.43|1.07|0.96|0.98|
> ||24|1.40|1.10|0.96|0.98|
> |u10|6|1.41|1.04|0.91|0.97|
> ||12|1.81|1.23|0.89|0.95|
> ||18|1.97|1.40|0.88|0.94|
> ||24|2.01|1.57|0.87|0.92|
> |v10|6|1.53|1.09|0.92|0.96|
> ||12|1.81|1.28|0.89|0.95
> ||18|1.96|1.45|0.88|0.94|
> ||24|2.04|1.61|0.86|0.92|
>
>
> > The clarity and text needs improvements.
>
> Thanks for this feedback; we will work on improving the clarity and text.
>
> ## **Questions**
>
> > I have hard time understanding the v0 estimation. First, a wave equation is introduced in general terms, and then a CNN appears. These two are not connected together, so I’m left wondering what do we do with the waves, and how do they relate to the CNN. Furthermore, the wave equation is poorly motivated, and it seems disconnected from the advection equation 3. To me this looks like a discrepancy: the system follows advection, but initial state assumes different kind of physics (ie. laplacians appear out of nowhere). Surely the initial state estimation needs to respect the chosen ODE model, and not introduce some physics effects that are not part of the ODE.
>
> Apologies for any confusion. To train the neural ODE effectively, it is crucial to accurately estimate the initial state, $v_0$. We believe that the inaccurate estimation of $v_0$ using $\frac{\Delta u}{\Delta t}$ in ClimODE leads to suboptimal results. **To address this, we introduce the wave equation under the assumption of incompressible flow, which relates $\frac{\partial u}{\partial t}$ to the spatial derivatives $(\frac{\partial u}{\partial x}, \frac{\partial u}{\partial y})$.** Although the wave equation involves second-order derivatives, **neural networks are capable of approximating both differentiation and integration processes.** Thus, by feeding $(\frac{\partial u}{\partial x}, \frac{\partial u}{\partial y})$ into the network, we can better model $\frac{\partial u}{\partial t}$. Regarding the CNN structure, **it was chosen due to its rapid convergence properties (will be clarified in later response)**, which help stabilize the training of the ODE and prevent numerical blow-up.

---

> > ### Author Response · Authors · 2024-11-18
> > **Response Continued**
> >
> > > The arguments about spatial vs temporal resolution are not convincing. Low temporal sampling rate does not necessarily mean that the temporal resolution is low: this depends on how quickly the process varies. It also feels misguided to say that spatial resolution is 100x higher than temporal resolution: this is only true if you take 32*64, and I don’t think you should do this. Since the (t,x,y) axes have similar ranges (24,32,64), I would argue that there is no resolution gap in the sampling rate between space and time.
> >
> > While the ranges of the temporal and spatial axes may appear similar (24, 32, 64), the key difference lies in the underlying periodicity: the temporal period is 24 hours, while the spatial period spans the entire Earth. In this context, we are comparing $1/24$ (time) vs. $1/(32 \times 64)$ (space). Of course, the dimensional units of time (hours) and space (km) are not directly comparable, making it challenging to logically determine the "ideal" resolution for both time and space.
> >
> > Therefore, we rely on experiments to validate our approach. In our neural ODE, we need to estimate the velocity, which can either be done using temporal differences ($\Delta t$) or spatially through a wave function. Initially, we could not be sure which approach would yield better results. **However, we found that the wave-based spatial estimation provided a clear improvement.** Interestingly, when we added more temporal information, we observed negative results (in Figure 3), reinforcing our hypothesis that the temporal resolution might be insufficient. **This suggests that time-based velocity estimation is less effective due to the inadequate temporal sampling frequency.**
> >
> > To further support this point, we supplement our experiments with precipitation forecasting tasks using the SEVIR dataset [1], which features a temporal resolution of **5 minutes** (significantly shorter than the 1-hour intervals in our primary experiments). **Table C** below demonstrates that when the temporal sampling is sufficiently dense, using $\frac{\partial u}{\partial t}$ instead of $\nabla u$ for velocity estimation becomes viable. **The results indicate that dense temporal resolution mitigates the disadvantages of temporal derivative approximation.**
> >
> > **Table C.** WeatherODE result on SEVIR dataset
> > ||CSI $\uparrow$|CSI_pool4 $\uparrow$|CSI_pool16 $\uparrow$|
> > |-|-|-|-|
> > |WeatherODE ($f_v(\frac{\Delta u}{\Delta t})$)|0.1953|0.2031|0.2258|
> > |WeatherODE ($f_v(\nabla u)$)|0.1926|0.2045|0.2290|
> >
> > (CSI is **Critical Success Index**)
> >
> > [1] SEVIR: A storm event imagery dataset for deep learning applications in radar and satellite meteorology.
> >
> > > couldn’t follow the CNN/ViT convergence arguments. I’m not sure what convergence even means here (of training..?, of rollouts..?). I think the paper is arguing that training CNNs is somehow instable, and thus ViT’s have to be used. This sounds implausible, and a poor motivation for choosing how the dynamics should evolve. Surely the networks need to be chosen such that they respect some physical system properties (eg. feature locality/globality).
> >
> > **By "convergence", we are referring to the convergence speed — that is, how quickly the model reaches a good solution during training.** This is not about the stability of training, but rather how fast the model can find an effective solution. CNNs, with their inductive biases like locality, translation invariance, and weight sharing, efficiently capture image features, allowing them to converge faster [2]. In contrast, ViTs, due to the self-attention mechanism, lack these biases and therefore converge more slowly.
> >
> > We leverage this difference by using CNNs, which converge quickly, to model the initial velocity $v_0$ and ViTs, which converge more slowly, to model the neural ODE. **This strategy helps mitigate the numerical blow-up caused by inaccurate $v_0$ estimation.** As shown in rows 5-8 of Table 3, using CNNs to model the neural ODE leads to numerical instability and blow-up within the first epoch of training due to the faster convergence of CNNs.
> >
> > [2] Role of Locality and Weight Sharing in Image-Based Tasks: A Sample Complexity Separation between CNNs, LCNs, and FCNs

---

> > > ### Author Response · Authors · 2024-11-18
> > > **Response Continued**
> > >
> > > > I don’t understand the source model. It takes as input all 1…N simulated states u(t_n), but one needs the source states s(t_n) to do this simulation. This has to be a mistake: I assume that we instead take as inputs u(t_1 : t_n), so that the history grows along the ODE rollouts. How do you handle set inputs and set outputs? I don’t think a 3D CNN supports sets as inputs or outputs [maybe these are not sets, but just tensors of fixed size..].
> > >
> > > Let us provide a more detailed explanation. Excluding the batch size dimension, the inputs to the Neural ODE are $u \in [K, H, W]$ and $v \in [2K, H, W]$, where $K$ is the number of input variables, and $v$ has $2K$ dimensionality because it represents velocity in two directions. The Neural ODE then outputs $\hat{u} \in [t, K, H, W]$ and $\hat{v} \in [t, 2K, H, W]$, where $t$ corresponds to the predefined number of ODE steps, which we set to match the lead time in hours. Each point along the $t$-dimension represents a one-hour forecast.
> > >
> > > For the source model, our goal is to learn the source term for $\hat{u} \in [t, K, H, W]$. We consider two approaches:
> > >
> > > 1. Treat $t$ into the batch dimension and use a 2D CNN to learn the source for all $t$-steps independently. However, this approach loses temporal information across adjacent time steps.
> > > 2. Use a 3D CNN, where the $t$-dimension is incorporated into the convolution kernel, allowing the model to capture relationships across adjacent time steps.
> > >
> > > As noted in our equations, the source term can also be learned at each iterative step within the Neural ODE. **However, our experiments show that this approach is highly sensitive to the cumulative errors introduced during ODE rollouts due to physical approximation inaccuracies.** Therefore, we instead choose to model the source using the outputs of all ODE steps collectively.
> > >
> > > We hope this clarifies the implementation and our design choices.
> > >
> > > > At sec 3.5. I’m again confused what do you mean by “convergence”.
> > >
> > > Explained in previous questions.
> > >
> > > > Sec 3.5. claims that advection dynamics involves long-range dependencies. I’m not sure I agree, and would even argue the opposite. PDE’s are by nature local models: why would the weather in new york affect the infinitesimal change of weather in london? There is no connection between them. Can you elaborate?
> > >
> > > The reviewer is correct that, in a strict sense, advection dynamics is local, as the weather in New York does not directly influence the weather in London. The advection equation itself primarily captures local dynamics. However, the issue arises when modeling these local dynamics over large regions. **If we only rely on local modeling, small errors accumulate over time, leading to significant inaccuracies.** In this context, CNNs with small receptive fields struggle to adjust for these cumulative errors because they cannot "see" the broader picture.
> > >
> > > On the other hand, the attention mechanism in transformers provides a global perspective. **It allows the model to learn that small errors in London might be overfitting, while larger errors in New York could signal a more systemic issue across the entire region.** Thus, the model adjusts parameters globally, ensuring consistency across all regions rather than focusing solely on local variances.
> > >
> > > We acknowledge that "long-range dependencies" might be misleading in this context, and "long-range consistency" might indeed be a more appropriate term to describe the global nature of the modeling process.

---

> > > > ### Author Response · Authors · 2024-11-18
> > > > **Response Continued**
> > > >
> > > > > Sec 3.6. claims that earlier methods only train against final state, and ignore intermediate terms. I’m surprised by this statement. ClimODE trains with all intermediate points, as does the original neural ODE, and every other neural ODE/PDE model I’ve seen. Can you provide examples of models where this happens? Calling this “multi-task” learning is also wrong: intermediate points in an ODE are not different “tasks” (they are not even a single repeated task, since it’s not a “task” in the first place).
> > > >
> > > > Apologies for the confusion. When we refer to "ignoring intermediate terms," we are specifically **comparing ODE-based models with non-ODE-based deep learning approaches, such as Pangu, FourCastNet, and ClimaX.** These models typically train a separate model for each lead time, learning the relationship for a specific $\Delta t$ (lead time), and rely on rolling predictions to handle different lead times.
> > > >
> > > > Regarding the term "multi-task learning," we should clarify that we consider predictions for different lead times as fundamentally distinct tasks in non-ODE models. **For example, a model predicting 12-hour temperatures may focus on the large temperature difference between morning and evening, while a 24-hour model would predict near-identical temperatures at the same time on consecutive days.** Non-ODE models cannot learn the varying mappings for different lead times within a single model. This is why models like Pangu and FourCastNet require separate models or rolling predictions for each lead time.
> > > >
> > > > In contrast, ODE-based models can learn continuous dependencies across time steps, rather than just mapping inputs to outputs at two distinct time points. **This highlights the advantage of ODEs in capturing the continuous temporal dynamics of weather forecasting.**
> > > >
> > > > > The benchmarks use 48 weather variables, but only evaluate 5 of them. The benchmarks also take ClimODE results as-is from the paper at least in Table 2. The results are then unfair to ClimODE: the ClimODE is using only 5 variables worth of information, while weatherODE is using 48 variables. The paper needs compare apple to apples by either running weatherODE results using only 5 variables, or running ClimODE with 48 variables.
> > > >
> > > > Running WeatherODE results using only 5 variables see **Table B**. Running ClimODE with 48 variables see the first column of **Table A** (compare it with Table 1's ClimODE result). Both demonstrate WeatherODE is superior to ClimODE.
> > > >
> > > > ---
> > > >
> > > > We hope Reviewer o4jc finds these clarifications helpful and appreciates the reasoning behind our design choices. Thanks again for valuable feedback.

---

> > > > > ### Comment · Reviewer_o4jc · 2024-11-23
> > > > >
> > > > > Thanks for the response. I still have few questions.
> > > > >
> > > > > I don't think I can follow the v0 estimation motivation. The PDE model is following advection equation assumption, while the v0 is estimated using wave equation assumptions. As far as I can see, these are incompatible with each other. You can't use wave equation to estimate v0 if the underlying PDE system does not follow wave equation. The wave equation might help the v0 estimation, but in that one needs to show why it helps (preferably with a theoretical argument). Can you still try clarify the situation?
> > > > >
> > > > > Furthermore, I find the way the CNN and PDE and wave equations all play together confusing. I would like to see a mathematical presentation of how all of these come together.
> > > > >
> > > > > I also couldn't follow the source estimation explanation in the response. The equations 1-3 all imply that the source is part of the time derivative, and is incrementally added to the state change. However, sec 3.3. and sec 3.4. say that the source is estimated as a set output of a set-input neural network, where apparently the source terms are estimated based on future states.
> > > > >
> > > > > I interpret the response such that the source is actually added after the PDE rollout as a correcting term, and it's not actually part of the time derivative function. That is, I think you run the PDE solver using only the advection term and without source term, and add the sources as kind of external corrections afterwards. Can you clarify if this is indeed the case? If this is the case, then eqs 1-3 are wrong. The paper needs to be mathematically precise and unambiguous to avoid this type of concerns.

---

> > > > > > ### Author Response · Authors · 2024-11-24
> > > > > > **Response**
> > > > > >
> > > > > > Thank you for your detailed follow-up. Below, we address the questions regarding the wave equation, CNNs, and the source term implementation in our neural ODE framework.
> > > > > >
> > > > > > #### 1. **Wave Equation and Advection Equation**
> > > > > > The advection equation is a specific case of the wave equation. All solutions of the advection equation satisfy the wave equation, as shown below:
> > > > > > $
> > > > > > \frac{\partial^2 u}{\partial t^2} - c^2 \frac{\partial^2 u}{\partial x^2} = \left( \frac{\partial u}{\partial t} + c \frac{\partial u}{\partial x} \right) \left( \frac{\partial u}{\partial t} - c \frac{\partial u}{\partial x} \right) u = 0.
> > > > > > $
> > > > > >
> > > > > > If $\frac{\partial u}{\partial t} + c \frac{\partial u}{\partial x}=0$, then $u$ is a solution of the wave equation. This relationship allows us to leverage the wave equation as a **weaker, more general bias** to estimate $v_0$. While the system follows the advection equation, the wave equation provides flexibility in incorporating spatial gradients ($\nabla u$) to approximate temporal derivatives ($\frac{\partial u}{\partial t}$), helping address coarse temporal resolutions.
> > > > > >
> > > > > > #### 2. **Role of CNNs**
> > > > > > The choice of CNNs is independent of the governing equations. We use CNNs primarily for their computational efficiency and their universal approximation capability, which allows them to approximate any function, including those derived from PDEs. Their role is simply to model the relationships learned from the data, regardless of whether the governing dynamics are based on the advection or wave equations.
> > > > > >
> > > > > > #### 3. **Neural ODE and Source Term**
> > > > > > To elaborate, $s(t)$ in Equations (1)-(3) is not an independent variable but a function of existing variables ($u, v, t, \dots$). Since $s(t)$ is entirely learned through the source network, both its value and gradient are unconstrained by physical assumptions and optimized directly. This approach allows $s(t)$ to be included as a differentiable term without explicitly embedding it into the advection dynamics during gradient calculations.
> > > > > >
> > > > > > We acknowledge that the exact variables $s$ depends on are not fully known—it could involve additional unobserved factors beyond those in our paper. However, in practice, our source network uses all available input variables as input, ensuring that the model can learn any potential dependencies. That‘s why we didn't put $s(t)$ as $s(u,v,t)$ in the first place because it could involve other variables. To reduce ambiguity, we can revise the notation in future versions to $s(u, v, t, \dots)$ to better reflect this dependency.  Does this align with your interpretation? If not, we’d be happy to clarify further. Thank you again for your valuable feedback.
> > > > > >
> > > > > > We hope this clarification resolves the concerns and highlights the mathematical consistency of our framework. Thank you again for your thoughtful questions and feedback.

---

> > > > > > > ### Comment · Reviewer_o4jc · 2024-11-24
> > > > > > >
> > > > > > > On wave. I'm not sure I understand. I don't see how the advection is related to the triple-equality equation given, or what this new equation even is. Advection has velocity field, while this new equation here doesn't seem to have. I really can't connect these two together. And even if we can connect these, it seems that one needs to make some drastic assumption of u_t = u_x, which I'm not sure if we are doing. I would recommend the authors to give a wider mathematical exposition to the model assumptions. Even if this is trivial for the authors, one needs to make all the the underlying physics assumptions precise, and make their connection to ML precise as well.
> > > > > > >
> > > > > > > I also don't really understand the source term still. I think now that the source term is a CNN over the (3K,H,W) tensor of (u,v) at one single timepoint. This however clashes with the sec 3.4. where the source is instead given as input the entire (N,3K,H,W) tensor of states at all timepoints. This is obviously impossible, since in the ODE/PDE forward unrolling only knows the state at the current (single) timepoint. I can't make sense of the system, and would appreciate a mathematically rigorous and complete description of the model at this stage.
> > > > > > >
> > > > > > > In the case the above concerns are not resolved, the paper is suffering from either substantial modelling or presentation issues. These would warrant changing my score to negative.

---

> > > > > > > > ### Author Response · Authors · 2024-11-25
> > > > > > > > **Response**
> > > > > > > >
> > > > > > > > Here is further explanations:
> > > > > > > >
> > > > > > > > ### **Source Network**
> > > > > > > >
> > > > > > > > Regarding the source network, allow us to provide further explanation. We're almost there—the only missing part is the feature generation trick. You are absolutely right, the only information we have is the observed state, and you can already see the source network can be a function of the observed state. Our next step is to enable the source network to engage in joint learning with a Neural ODE. By solving the Neural ODE, we can derive $N$ future predicted states from a single observed state. These $N$ predicted states form a temporal tensor of shape $(N, 3K, H, W)$, which serves as input to the source network. This resolves the concern about Section 3.4: the source term is indeed modeled based on the entire $(N, 3K, H, W)$ tensor, consistent with the Neural ODE integration process. The $N$ states are generated as part of the Neural ODE's trajectory prediction and are not directly observed but predicted states used in a single inference step.
> > > > > > > >
> > > > > > > > Essentially, we're employing a feature generation technique where one state generates $N$ features, which then serve as input for learning. This process is akin to using a single word (one-hot encoded) to create an $N$-dimensional embedding vector for subsequent calculations. The difference is that while word embeddings use a codebook for learning these embeddings, our approach uses a Neural ODE to generate new $N$ states. Moreover, this feature generation trick enables us to predict $N$ future states from a single inference path, demonstrating an advantage over direct forecasting that requires $N$ separate models. We hope this explanation is clear and easy to follow.
> > > > > > > >
> > > > > > > > ---
> > > > > > > >
> > > > > > > > ### **Wave Equation**
> > > > > > > >
> > > > > > > > We adhered to the notation established in Chandrasekar A.'s *Numerical Methods for Atmospheric and Oceanic Sciences* [1]. Specifically, in Sections 1.6 and 1.7, the symbol $c$ represents wave speed, which is analogous to the velocity field $v$ in the advection equation, maintaining the same physical interpretation. This equivalence should have been articulated more explicitly. Furthermore, this notation convention, where $c$ denotes $v$ in the context of the advection equation, is also employed in Section 6. We apologize for any confusion caused by this choice of notation.
> > > > > > > >
> > > > > > > > Building on this equivalence, the triple-equality equation provided earlier can be understood as a one-dimensional derivation. Extending it to two dimensions results in:
> > > > > > > >
> > > > > > > > $
> > > > > > > > \frac{\partial^2 u}{\partial t^2} - v^2 (\nabla u)^2 = \left( \frac{\partial u}{\partial t} + v \cdot \nabla u \right) \left( \frac{\partial u}{\partial t} - v \cdot \nabla u \right) u = 0.
> > > > > > > > $
> > > > > > > >
> > > > > > > > Here, the term $\frac{\partial u}{\partial t} + v \cdot \nabla u = 0$ should look familiar—it corresponds to the advection equation without the compression term ($u \nabla \cdot v$). This derivation underscores our point that any solution to the advection equation is inherently a solution to the wave equation. The absence of the compression term follows our approximation that, for a large area, the compression term can be neglected to ease the computation, as discussed in Section 3.2.
> > > > > > > >
> > > > > > > > [1] Chandrasekar, A., 2022. Numerical Methods for Atmospheric and Oceanic Sciences. Cambridge University Press.

---

> > > > > > > > > ### Comment · Reviewer_o4jc · 2024-11-26
> > > > > > > > >
> > > > > > > > > Thanks for the response. Unfortunately I can't make sense of either explanations. The treatment of source is still vague, and I can't decipher the triple equation or wave equation here, or connect them to the paper. The triple equation has a strange vector square, and some kind of compression-free double advection term. Not sure where this comes from.
> > > > > > > > >
> > > > > > > > > I believe the eqs 1-3 are incorrect in the paper since source term gets as input future states, and thus source can't be part of the time derivative. I also believe that the wave equation is unjustifed for estimating the initial velocity of advective systems. Our discussion has done little to clarify these issues, and the responses have not been convincing or rigorous. Perhaps I'm wrong here and the model and math is all fine, but even in that case the presentation would be inadequate. In the current form the paper is not ready for publication, and needs to be rejected.  I'm decreasing my score to rejection.

---

### Official Review · Reviewer_dcuz · 2024-10-20

**Soundness:** 2
**Presentation:** 3
**Contribution:** 2
**Rating:** 6
**Confidence:** 4

**Summary:**

The study proposes a neural ODE approach to data-driven NWP by using three different neural nets with three different rates of convergence to solve for three different terms in the system of ODEs obtained by applying the method of lines to the advection continuity equation. The three models correspond to (a) solving for the initial velocity estimate using the wave equation, (b) an advection model to compute the tendency of the velocity, and (c) to estimate the source term in the momentum equation. Respectively, a 2D CNN, a vision transformer and a 3D CNN is used to estimate the three terms which are then used to march the variables in time.

The model is compared with multiple other AI NWP models and this approach is found to substantialy reduce prediction errors in both global and regional contexts mst likely by reducing the errors in the estimation of the time derivative of velocities which might be quite erroneous due to a large (hourly) time step.

The ultimate approach is to the create a "sandwich"  physics driven ODE which uses the three architectures together to predict the future state of the atmosphere.

**Strengths:**

The experimental setup is robust and it is nice to see the significant improvements in performance across all variables for the neural ODE sandwich architecture. The paper is well written and the hybrid approach in the paper could likely be valuable to the AI NWP community.

I like the paper for its skillful approach of blending classical PDE theory with AI to improve errors in atmospheric state forecasting.

**Weaknesses:**

1) I don't think ClimaX is a good baseline for such comparisons, as it was itself trained on a wide range of CMIP6 models which are (a) not really tuned for weather forecasting and (b) due to model design have multiple biases. Finetuning can only fix so many of them. Thus, I would not put too much weight into how the WeatherODE performs against ClimaX as it is a weak baseline to begin with. Similarly, I would recommend comparing the method with the newer version of FourCastNet based on the SFNO architecture which is more stable for short and long term rollouts and is more topology-aware.

2) A minor point but the paper tends to over-cite at places: the Evans (2022) reference, the Vaswani et al reference and the Biswas et al. (2013) reference are totally unnecessary. Vaswani et al. has not even been cited in the correct context as it has not realtion to weather forecasting. Similarly on Line 37-38.

3) The figures' text could be increased. It is tough to interpret Figures 1 and 4 in print.

4) It would have been interesting to see the time complexity of the sandwich model proposed in the study and how it compares with other models. Current AI weather forecasting models have been a bit over-glorified in a sense that they ignore baroclinic motions in the atmosphere and use data on very few vertical levels, and then claim to be as good as traditional NWP models - which are way more versatile thab AI NWPs in the problems they can be used to solve. Understandably, the only advantage then is the computational speed and the reduction in operational cost that they offer. Therefore, as more traditional numerical are embedded into pure data-driven architectures, it would be interesting to see the effect on more complicated hybrid architectures on the run speed.

5) If the refined horizontal resolution is the key here (as it allows computing spatial gradients accurately), they can simply using higher spatial resolution training data (like 25 km x 25 km ERA5 data) lead to similar improvements in performance?

6) One worry is that the achitecture is becoming too combersome to be practical: a unified model structure for past AI NWP models is one of the key cornerstones of its appeal. Traditional NWP model provide forecasts at a 9 km resolution. If the same models were to be run at 5.625 deg resolution, one can expect similar order run times between AI weather emulators and NWP models, ultimately leading one to question the central point of these models. So, having multiple ML models in sequence, such as those proposed in this study, can increase the complexity of the problem to the point that one begins to question the novelty of this approach. What would have been great could be to train one single neural net on atmospheric data sampled at a high frquency and use that for more accurate initializations.

7) If i undestand correctly, the study employs a two dimensional equation. Have the authors considered using a three dimensional equation instead which considers the vertical verlocity into account as well? This could be important especially for tropical predictability ad thermodynamical processes like dry and moist convection evolve over sub-hourly timescales and can introduce notable errors into the equation. This could also affect longer lead time rollouts of the model. Or, if I undertand it correctly, the vertical velocity is simply treated within the source term (which might not be the best approach).

8) Source term: If I understand correctly, and I could be wrong, the source model tend to learn all the other forcing terms of the advection equation. In the context of ERA5, this would not just contain other forcings, but also data assimilation errors. Since the DA errors are not bery systematics in nature, how can errors in predicting the terms influence the u_t+1 prediction obtained from the neural ODEs?

**Questions:**

Questions are combined with weaknesses.

---

> ### Author Response · Authors · 2024-11-18
> **Response**
>
> We sincerely thank Reviewer dcuz for thoughtful insights and recognition of our effort to integrate classical PDE theory with AI to enhance weather forecasting. Their championing of our work truly inspires us to pursue this research direction. Below, we address each point raised, offering clarifications and detailed responses.
>
> > I don't think ClimaX is a good baseline for such comparisons, as it was itself trained on a wide range of CMIP6 models which are (a) not really tuned for weather forecasting and (b) due to model design have multiple biases. Finetuning can only fix so many of them. Thus, I would not put too much weight into how the WeatherODE performs against ClimaX as it is a weak baseline to begin with. Similarly, I would recommend comparing the method with the newer version of FourCastNet based on the SFNO architecture which is more stable for short and long term rollouts and is more topology-aware.
>
> While it is true that ClimaX is pretrained on CMIP6 models and finetuned on ERA5, it has demonstrated strong results at both 5.625° and 1.4° resolutions. **We compared WeatherODE with ClimaX primarily because ClimODE's comparison to ClimaX involved a non-pretrained ClimaX trained on a subset of ERA5 data. Thus, we felt it was appropriate to compare our method with the original ClimaX.**
>
> As for FourCastNet based on the SFNO architecture, their model was trained on 0.25° data, and the specific training process is not publicly available. However, from Figure 6 in their paper [1], their 0–24h performance is slightly below IFS, similar to the gap we observe between WeatherODE and IFS. **This suggests that WeatherODE is comparable in performance to FourCastNet based on SFNO.**
>
> Our decision to train on 5.625° data was constrained by computational resources, as training on 0.25° data would require thousands of GPUs, which we do not have access to, unlike larger companies such as Google or NVIDIA. **Nevertheless, the scaling laws in deep learning—where larger datasets and models yield better results—have been widely validated in CV and NLP.** We anticipate that WeatherODE, with access to larger data and model sizes, could surpass IFS in performance.
>
> [1] Spherical Fourier Neural Operators: Learning Stable Dynamics on the Sphere
>
> > A minor point but the paper tends to over-cite at places: the Evans (2022) reference, the Vaswani et al reference and the Biswas et al. (2013) reference are totally unnecessary. Vaswani et al. has not even been cited in the correct context as it has not realtion to weather forecasting. Similarly on Line 37-38.
>
> Thank you for pointing this out. The citations were intended to reference the original works where PDEs, the Euler method, and Transformers were first introduced. We agree that they are not directly relevant to weather forecasting. We will revise these citations in the updated version of the paper.
>
> > The figures' text could be increased. It is tough to interpret Figures 1 and 4 in print.
>
> Thank you for pointing this out. In the revised version, we will carefully review the text size of all figures and adjust them to ensure they are easily readable in print.
>
> > It would have been interesting to see the time complexity of the sandwich model proposed in the study and how it compares with other models. Current AI weather forecasting models have been a bit over-glorified in a sense that they ignore baroclinic motions in the atmosphere and use data on very few vertical levels, and then claim to be as good as traditional NWP models - which are way more versatile thab AI NWPs in the problems they can be used to solve. Understandably, the only advantage then is the computational speed and the reduction in operational cost that they offer. Therefore, as more traditional numerical are embedded into pure data-driven architectures, it would be interesting to see the effect on more complicated hybrid architectures on the run speed.
>
> Models that combine deep learning with physical constraints, such as WeatherODE, ClimODE, and NeuralGCM, generally have longer inference times per step compared to purely data-driven models like FourCastNet and Pangu. **However, physics-constrained models often offer greater flexibility, as they can predict multiple time steps in a single inference** (as WeatherODE* does).
>
> In contrast, purely data-driven models typically train a separate model for each lead time and rely on rolling strategies for multi-step forecasts (e.g., using a 1-hour model twice to predict 2 hours ahead). When predicting multiple consecutive lead times, physics-constrained models can thus have an advantage in efficiency, as they avoid the repeated inference required by rolling strategies in purely deep learning approaches.

---

> ### Author Response · Authors · 2024-11-18
> **Response Continued**
>
> > If the refined horizontal resolution is the key here (as it allows computing spatial gradients accurately), they can simply using higher spatial resolution training data (like 25 km x 25 km ERA5 data) lead to similar improvements in performance?
>
> We agree with this observation. Increasing the resolution to 1.4° or even 0.25° would likely improve performance. **Higher resolution not only allows for more accurate computation of spatial gradients, as you noted, but also benefits from the performance gains associated with the deep learning scaling laws.** We hope that our work can be further validated in the future by organizations with greater resources on larger-scale and finer-resolution datasets.
>
> > One worry is that the achitecture is becoming too combersome to be practical: a unified model structure for past AI NWP models is one of the key cornerstones of its appeal. Traditional NWP model provide forecasts at a 9 km resolution. If the same models were to be run at 5.625 deg resolution, one can expect similar order run times between AI weather emulators and NWP models, ultimately leading one to question the central point of these models. So, having multiple ML models in sequence, such as those proposed in this study, can increase the complexity of the problem to the point that one begins to question the novelty of this approach. What would have been great could be to train one single neural net on atmospheric data sampled at a high frquency and use that for more accurate initializations.
>
> Thank you for the excellent question. Our work explores a different perspective, leaning towards data-driven learning rather than adhering strictly to the constraints of traditional NWP models. **While NWP provides solutions based on human-defined ODEs, we believe these models may not fully capture the complexities or imperfections of the real world.** Simply solving these equations faster and more accurately may not necessarily lead to better predictions.
>
> Completely data-driven learning, however, poses its own challenges. It requires enormous amounts of data, far beyond what is currently available, to replicate centuries of accumulated physical knowledge. Rare weather phenomena, for instance, are difficult for ML models to learn from limited examples.
>
> **We aim to strike a balance by integrating minimal but critical physical knowledge, guided by Occam’s Razor: "We consider it a good principle to explain the phenomena by the simplest hypothesis possible."** This approach maintains enough flexibility for the model to learn autonomously from data while incorporating essential constraints from physics. While the path you suggest—relying on NWP with sufficient computational resources and ideal initial conditions—might work if our understanding of weather dynamics were complete, we believe a hybrid approach, where ML leads with NWP offering key guidance, provides a more robust path to accurate forecasting.
>
> > If i undestand correctly, the study employs a two dimensional equation. Have the authors considered using a three dimensional equation instead which considers the vertical verlocity into account as well? This could be important especially for tropical predictability ad thermodynamical processes like dry and moist convection evolve over sub-hourly timescales and can introduce notable errors into the equation. This could also affect longer lead time rollouts of the model. Or, if I undertand it correctly, the vertical velocity is simply treated within the source term (which might not be the best approach).
>
> Thank you for raising this excellent point. This study primarily focuses on a two-dimensional equation. We acknowledge the importance of incorporating vertical velocity, especially for capturing tropical predictability and thermodynamic processes.
>
> We are actively exploring extensions to three-dimensional modeling. **For example, ERA5 data includes variables across multiple pressure levels, which could serve as a basis for expanding the advection equation to three dimensions by introducing a vertical velocity component $v_z$.** This is a promising direction for future research, and we anticipate that incorporating 3D dynamics could further enhance the model’s accuracy and generalizability.

---

> > ### Author Response · Authors · 2024-11-18
> > **Response Continued**
> >
> > > Source term: If I understand correctly, and I could be wrong, the source model tend to learn all the other forcing terms of the advection equation. In the context of ERA5, this would not just contain other forcings, but also data assimilation errors. Since the DA errors are not bery systematics in nature, how can errors in predicting the terms influence the u_t+1 prediction obtained from the neural ODEs?
> >
> > Thank you for the excellent question. Reanalysis data (ERA5) is generally considered the closest approximation to the true state of the atmosphere, as it represents an optimal estimate obtained by balancing observation and NWP error covariances. While ERA5 does contain noise, **this is a common challenge in many deep learning tasks,** such as time-series forecasting, stock prediction, and video prediction, where input data often includes random noise. **Deep learning models aim to overcome this by learning robust representations that are less sensitive to noise.** Techniques like data augmentation or intentional perturbations are widely used in image classification to help models generalize in noisy environments. To further address your concern, we conduct additional experiments to evaluate the effect of perturbation noise. Specifically, for the 24-hour WeatherODE* model, we perturbed the input five times, generated the corresponding outputs, and averaged them to create an ensemble forecast. **Table A** summarizes the results, **showing that the ensemble method slightly improves performance compared to the original model. This finding reinforces our earlier analysis, demonstrating the model's robustness even under input noise.**
> >
> > In our case, the source model is not learning noise in the traditional sense but rather the systematic bias between the true (unknown) physical model and the simplified advection equation we define. While ERA5's inherent noise may introduce some overfitting to irrelevant patterns, this issue was not directly addressed in this work. However, approaches such as introducing perturbation noise during training or augmenting the data could help the model learn more robust representations. This is an exciting direction for future research, and we hope to explore it further.
> >
> >
> > **Table A.** Ensemble experiment of WeatherODE*
> > |||RMSE $\downarrow$|||ACC $\uparrow$|||
> > |-|-|-|-|-|-|-|-|
> > |Variable|Lead Time|WeatherODE* |WeatherODE* (Ensemble)|WeatherODE*|WeatherODE* (Ensemble)|
> > |z500|6|56.3|56.5|1.00|1.00|
> > ||12|73.3|72.8|1.00|1.00|
> > ||18|91.9|90.6|1.00|1.00|
> > ||24|114.5|113.9|1.00|1.00|
> > |t850|6|0.76|0.75|0.99|0.99|
> > ||12|0.88|0.87|0.98|0.99|
> > ||18|0.95|0.95|0.98|0.98|
> > ||24|1.04|1.03|0.98|0.98|
> > |t2m|6|0.78|0.77|0.99|0.99|
> > ||12|0.89|0.89|0.98|0.98|
> > ||18|0.95|0.94|0.98|0.98|
> > ||24|0.98|0.98|0.98|0.98|
> > |u10|6|0.88|0.87|0.98|0.98|
> > ||12|1.00|0.99|0.97|0.97|
> > ||18|1.13|1.11|0.96|0.96|
> > ||24|1.26|1.24|0.95|0.95|
> > |v10|6|0.90|0.90|0.98|0.98|
> > ||12|1.04|1.03|0.97|0.97|
> > ||18|1.16|1.14|0.96|0.96|
> > ||24|1.29|1.27|0.95|0.95|
> >
> > ---
> >
> > We hope Reviewer dcuz finds these clarifications helpful and appreciates the reasoning behind our design choices. Thanks again for valuable feedback.

---

> ### Comment · Reviewer_dcuz · 2024-11-25
> **I am not very convinced with some of the responses and having more time to digest the methodology, modify my score a bit - part 1**
>
> I appreciate the authors' effort in preparing the response, but having had more time to think about the paper and digesting the underlying methodology better now, I reduce my score a bit.
>
> 1) ClimaX operating at 5.625 degree and 1.4 degree resolution is akin to a toy model for weather prediction at best. I reemphasize that predicting the large-scale features at 5.625 degrees with reasonable accuracy is a much easier problem than learning the actual multi-scale dyanamics from data - simplified models like Quasi-geostrophic equations can be used at these resolutions to get reasonbale forecasts as well over 12-24 hr timescales, while taking a fraction of the compute cost at NWP.
>
> 2) Sure, the neural scaling laws can be invoked to surmise that the model will continue to scale with increased resolution, but I still do not see it being independent of the underlying reanalysis. The model will continue to be (a) more computationally intensive, (b) require newer and newer reanalysis datasets, and (c) based on current equations, still not have limited value for long term forecasts. Moreover, increasingly complex equations will be needed to ensure stable rollouts over longer periods - which connects to my next point.
>
> 3) Coriolis term? What about other physics on longer time scales? A heavy limitation of your approach is that you will eventually have to embed more complex equations into the hybrid architecture - at which point the architecture might converge to existing models like NeuralGCMs.
>
>
> 4) I am not convinced by the authors' response on the computational complexity. Of course, auto regressive rollouts will be more expensive than single-shot predictions, but if all one is getting from the (arguably more-complex) hybrid approach is marginal improvements over 6-24 hour timescales, that barely has any significant utility because pure-data driven models are relatively easier to train, and the auto-regressive rollout issue can be circumvented for medium-range timescales, as has recently been done with the weather and climate foundation model Prithvi wxc (if i understand correctly).

---

> ### Comment · Reviewer_dcuz · 2024-11-25
> **I am not very convinced with some of the responses and having more time to digest the methodology, modify my score a bit - part 2**
>
> Continued from above ...
>
> 5) "While NWP provides solutions based on human-defined ODEs, we believe these models may not fully capture the complexities or imperfections of the real world."
>
> I disagree with the authors here when they claim that NWP models can not capture the complexities of the real world but the data-driven methods can. I must remind them that the data-driven models (a) are trained on datasets created by these equations-driven traditional NWP models (along with observations) and (b) employ a simplified form of same equations for hybrid modeling. The only benefit AI-driven weather emulators have delivered on till date is speed. None of the models have systematically beaten IFS. Moreover, when integrated at a high-enough resolution, NWP models (without assimilation) can still be maintained physics-fidelity over longer periods of time than the state-of-the-art data driven models. It has become a fashion to bash traditional NWP models these days withouth appreciating the heavy amount of physics that goes into their design. Until these data-driven models show significant improvements over IFS after being trained on IFS input itself, alas, such claims about imperfections of the NWP models will remain baseless.
>
> This tradeoff between model complexity and performance and how it contributes towards interpretability is argued in more detail in [1] which provides a comprehensive discussion of the different approaches - purely data-driven to hybrid to pure-physics based, and, simple models to intermediate complexity models to comprehensive weather forecasting and climate prediction model. I highly recommend the authors to atleast acknowledge such tradeoffs between model complexity, realism, and accuracy in the final manuscript as limitations of their approach and connect it to [1] (and other studies).
>
> Also, a complete data-driven approach also leads to reduced interpretability of the model, as is again argued in [1] and multiple other studies. How do you think your approach embeds increased interpretability?
>
> References:
> [1] Laura A. Mansfield, Aman Gupta, Adam C. Burnett, Brian Green, Catherine Wilka, and Aditi Sheshadri. Updates on Model Hierarchies for Understanding and Simulating the Climate System: A Focus on Data-Informed Methods and Climate Change Impacts. Journal of Advances in Modeling Earth Systems, 15(10):e2023MS003715, 2023. ISSN 1942-2466. doi: 10.1029/2023MS003715.
>
>
> 6) Regarding ensemble averaging: no offense but a 0.01 or a 0.02 decrease in RMSE does not equate to much and should not be read much into. A better test would be around case-specific events. If you can show that ensemble methods lead to better predictability around, say, a cyclogenesis events, that would be something. Sorry if this sounds a bit rough, but, I think the AI weather prediction community needs to be reminded every now and then that this is not merely an engineering problem where comparing RMSEs will suffice. What is needed is a set of physically more rigrous tests that demonstrate that these models are physically robust and are capable of being used for meaningful physical analysis.

---

### Official Review · Reviewer_g7RP · 2024-10-25

**Soundness:** 2
**Presentation:** 3
**Contribution:** 2
**Rating:** 5
**Confidence:** 3

**Summary:**

This paper introduces WeatherODE, a novel weather forecasting model based on deep learning. Building on the recent ClimODE model, WeatherODE follows a similar methodology and models the evolution of atmospheric quantities through a learnable advection equation. In this framework, the authors propose new architectures for learning the advection equation. A CNN-based neural network is used to predict the initial velocity of the equation, leveraging insights from the wave equation, rather than relying on time discretization. Additionally, a Vision Transformer-based neural ODE is trained to model the evolution of velocity over time. Finally, a CNN-based neural network is introduced to learn the source term of the advection equation, aiming to reduce the propagation of numerical errors. The paper provides detailed information on the training process of this new architecture, along with extensive experiments comparing WeatherODE to several state-of-the-art baselines across global and regional weather forecasting tasks with varying lead times. Ablation studies on the model's architecture are also included.

**Strengths:**

The paper is well organized with a clear structure and informative figures and tables. The experimental work is extensive and technically well-documented.

Incorporating a strong physical bias into learning models is of great importance, especially for climate-related problems where the chaotic nature of underlying physical processes can cause the system to deviate from the training distribution. The inclusion of physical priors is essential for ensuring that the model generalizes well.

The proposed method outperforms the other models by a significant amount, in both global and regional forecasts.

The proposed method significantly outperforms the baselines, demonstrating improvements in both global and regional forecasting tasks.

Ablation studies, accompanied by illustrative explanations of the core contributions, are provided. The paper, in particular, demonstrates the effect of time discretization and how the proposed architecture mitigates this issue, offering comparisons to previous methods.

The discussion on the convergence speed of various learning models is first introduced qualitatively in Section 3.3 and then quantified in Section 5.3, which offers valuable insights into the architecture's design choices.

Although briefly mentioned, the paper also proposes a flexible inference model capable of producing forecasts for different lead times. The ability to make weather predictions at varying lead times is a significant advantage and a promising idea.

**Weaknesses:**

Despite the solid experimental results, I believe that the paper's explanation of the methodology lacks some clarity, context, and references. While the proposed architecture achieves strong performance, the scientific reasoning behind these improvements is somewhat incomplete.

First, the authors claim that all physical variables follow an advection equation, which is learned. This assumption, initially made in the ClimODE model, forms the foundation of the WeatherODE architecture and is said to provide a strong physical bias. It would be beneficial if the paper provided more physical context regarding the advection equation and why it serves as a good prior for learning atmospheric dynamics. From a physical perspective, what are the consequences of assuming that all variables follow this equation with a learned velocity field? Can the authors connect this to known atmospheric processes?


#### Section 3.1

There is some lack of clarity in the discussion of discretization between lines 192 and 204, particularly around Equation (3). It is unclear at which stage the ODE is being discretized. The problem of learning an ODE's flow using neural ODEs should, in theory, be independent of the discretization, as neural ODEs are designed to differentiate through the ODE in a solver-agnostic way.  Second, indices $t_0$ and $t_n$ seem to play the same role.  It would be clearer if a single notation were used for the initial condition, even if this involves only one discretization step, with the methodology then being extended to subsequent steps.

####  Section 3.2

This section focuses on predicting the initial velocity $v(t_0)$. The authors provide physical insight on the computation of the state derivative $\partial u/ \partial t$. However, the link between the two quantities $v(t_0)$ and $\partial u / \partial t$ is not explicitly stated. Yet it is one of the paper's main objectives to improve the computation of the $v$ as a function of the state $u$, without resorting to time discretization. Indeed, it is stated that $v$ is estimated from $u$ in the ClimODE paper using time discretization, but the estimation is never mentioned. As a result, the section presents a new method based on the wave equation to estimate state derivatives, but the connection with the ultimate problem of estimating the velocity is missing. This omission impairs the clarity of the paper, as this step is one of the core contribution.
I understand that the estimation method is framed as a variational inverse problem in the ClimODE paper, and that WeatherODE improves it by casting it as a learning problem instead. I believe that it should be stated in this paper as well for Section 3.2 to be relevant.

Additionally, the link between the wave equation and the advection equation is not clearly explained. The authors mention that the wave equation is commonly used in atmospheric dynamics, but they do not provide any references or detailed physical context. While the wave equation is indeed important in describing physical processes related to propagation, further clarification is needed regarding its purpose in this model.

If I understood correctly, the integrated wave equation (5) seems to show that the first-order derivative of the state $\partial u / \partial t (t_0)$ is linked to the state gradients $\nabla u(t_0)$, which motivates predicting the velocity field should as a function of the state gradients rather than the state itself. However, assuming that the wave equation holds, the involved spatial derivatives are of second order rather than first order, and are integrated over a past time interval rather than evaluated only at the current time $t_0$. In my opinion, and if I understood correctly, equation (5) may not be entirely appropriate for predicting $v(t_0)$ as described.

This concern is compounded by the results of Figure 3, where the performance gap between the proposed method using the wave equation and other approximators is quite small. Couldn't it be that the performance gap comes from learning the initial velocity from a neural network conditioned on non-discretized state values instead of solving an inverse problem with discretized state derivatives, rather than from the wave equation-informed predictive structure of the neural network?

Finally, the qualitative argument about spatial resolution at the end of this section is not particularly convincing, as it compares spatial resolution to time resolution in a way that seems dimensionally inconsistent. The statement on line 243,
``
the spatial domain is nearly 100 times denser than the temporal domain.
 ``

needs further clarification, as its implications are unclear.

**Questions:**

Neural ODEs are known to be computationally demanding, yet the paper does not address computational resources or the back-propagation method used to train the neural ODE. Since computational power can often be a limiting factor, it would be valuable to compare the computational times across the different architectures, particularly for the neural ODE versus other feedforward models.

The WeatherODE* model appears to be an interesting research direction, offering flexibility in generating forecasts at different lead times. However, there is limited explanation as to why WeatherODE is capable of such flexibility. Could the authors clarify the connection between the following sentence:

``
by modeling the atmosphere as a physics-driven continuous process anddesigning a time-dependent source network to account for errors at each time step, WeatherODE can capture information across all intermediate time points
``

 and the success of the 24-hour model WeatherODE*?

---

> ### Author Response · Authors · 2024-11-18
> **Response**
>
> We sincerely thank Reviewer g7RP for their detailed feedback and thoughtful comments, which demonstrate a deep understanding of our work. Below, we address each point raised with clarifications and detailed responses.
>
> ## **Weakness**
>
> > First, the authors claim that all physical variables follow an advection equation, which is learned. This assumption, initially made in the ClimODE model, forms the foundation of the WeatherODE architecture and is said to provide a strong physical bias. It would be beneficial if the paper provided more physical context regarding the advection equation and why it serves as a good prior for learning atmospheric dynamics. From a physical perspective, what are the consequences of assuming that all variables follow this equation with a learned velocity field? Can the authors connect this to known atmospheric processes?
>
> From a physical perspective, the foundation of our approach lies in fundamental conservation laws: conservation of mass, energy, and momentum. **The advection equation, specifically, is derived from the principle of mass conservation, which asserts that the inflow and outflow of mass or substances within a given spatial region remain balanced.** This makes it one of the most fundamental conservation laws, as energy and momentum conservation often involve more complex dissipative processes, whereas mass is relatively stable.
>
> While not all atmospheric variables are perfectly suited to be described by mass conservation, it provides a general and reasonable physical bias. **Importantly, the velocity field in our model is variable-specific, meaning each variable follows its own dynamics rather than adhering to a single universal motion.** The shared assumption is that these dynamics align with the essential process of mass conservation, particularly at a large regional scale.
>
> **Although the atmosphere is compressible on micro scales, our model operates over larger patches, where the assumption of approximate mass conservation within a region is reasonable.** This provides a flexible yet grounded physical context for the advection-based modeling framework, balancing simplicity with general applicability.
>
> >  There is some lack of clarity in the discussion of discretization between lines 192 and 204, particularly around Equation (3). It is unclear at which stage the ODE is being discretized. The problem of learning an ODE's flow using neural ODEs should, in theory, be independent of the discretization, as neural ODEs are designed to differentiate through the ODE in a solver-agnostic way. Second, indices t0 and tn seem to play the same role. It would be clearer if a single notation were used for the initial condition, even if this involves only one discretization step, with the methodology then being extended to subsequent steps.
>
> Allow me to clarify. In the context of Neural ODEs, the discretization referenced in our paper corresponds to the series of intermediate "adjoint states" [1] computed between the initial input and the final output. Specifically, the input to our ODE is $u \in [K, H, W]$ and $v \in [2K, H, W]$, where $K$ represents the number of input variables, and $v$ has $2K$ dimensions as it includes velocity components in two directions. The ODE outputs $\hat{u} \in [N, K, H, W]$ and $\hat{v} \in [N, 2K, H, W]$, where $N$ corresponds to the number of predefined ODE steps ($t_1, \dots, t_N$), which we set to align with the lead time in hours.
>
> **At each step of the ODE, the next state of $u$ is calculated using the current state of $u$ and $v$ via the advection equation, while the next state of $v$ is predicted by the neural network.** This design allows the ODE to incorporate supervision from ground truth at every intermediate time step, not just the final state.
>
> Regarding the indices $t_0$ and $t_n$, apologies for the confusion. In Equation (3), **the underbraces correspond to the components being modeled at different stages**.
> 1. **Initial velocity ($v(t_0)$)**: The initial velocity $v_0$, fed into the ODE, is modeled by the velocity network. In the term $\begin{bmatrix} u(t_n) \\ v(t_n) \end{bmatrix}$, $u(t_0)$ is known, while $v(t_0)$ needs to be inferred by the network.
> 2. **Advection ODE ($\dot{v}(t_n)$)**: The Neural ODE models the advection dynamics.
> 3. **Source term ($s(t_n)$)**: The source model predicts the source term $s(t_n)$.
>
> We hope this clarifies your concerns. In the revised version, we will elaborate on these points to ensure greater clarity.
>
> [1] Neural Ordinary Differential Equations

---

> > ### Author Response · Authors · 2024-11-18
> > **Response Continued**
> >
> > > This section focuses on predicting the initial velocity v(t0). The authors provide physical insight on the computation of the state derivative ∂u/∂t. However, the link between the two quantities v(t0) and ∂u/∂t is not explicitly stated. Yet it is one of the paper's main objectives to improve the computation of the v as a function of the state u, without resorting to time discretization. Indeed, it is stated that v is estimated from u in the ClimODE paper using time discretization, but the estimation is never mentioned. As a result, the section presents a new method based on the wave equation to estimate state derivatives, but the connection with the ultimate problem of estimating the velocity is missing. This omission impairs the clarity of the paper, as this step is one of the core contribution. I understand that the estimation method is framed as a variational inverse problem in the ClimODE paper, and that WeatherODE improves it by casting it as a learning problem instead. I believe that it should be stated in this paper as well for Section 3.2 to be relevant.
> >
> > Thank you for highlighting this point. The connection between $\frac{\partial u}{\partial t}$ and $v(t_0)$ is indeed discussed in ClimODE. ClimODE notes that solving Neural ODEs requires an accurate estimate of the initial velocity $v(t_0)$. To compute $v(t_0)$, we revisit the advection equation:  $\frac{\partial u}{\partial t} + v \cdot \nabla u + \nabla v \cdot \nabla v = 0.$ **At $t_0$, $u(t_0)$ is known (the input), which means $\nabla u(t_0)$ is also known. Thus, estimating $v(t_0)$ requires a good approximation of $\frac{\partial u}{\partial t}$.**
> >
> > ClimODE approximates $\frac{\partial u}{\partial t}$ using time discretization, specifically $\frac{\Delta u}{\Delta t}$, derived from past time steps. However, we argue that this approach introduces significant errors. **To address this, we leverage wave equation theory, which reveals that $\frac{\partial u}{\partial t}$ can also be estimated using $\frac{\partial u}{\partial x}$ and $\frac{\partial u}{\partial y}$.** By incorporating spatial derivatives, we achieve a more accurate estimate of $\frac{\partial u}{\partial t}$, leading to a better estimation of $v(t_0)$.
> >
> > We apologize for not making this connection more explicit in the paper. In the revised version, we will provide additional context from ClimODE to clarify the relationship between $\frac{\partial u}{\partial t}$ and $v(t_0)$ and explain how our method improves upon the previous approach.

---

> > > ### Author Response · Authors · 2024-11-18
> > > **Response Continued**
> > >
> > > > Additionally, the link between the wave equation and the advection equation is not clearly explained. The authors mention that the wave equation is commonly used in atmospheric dynamics, but they do not provide any references or detailed physical context. While the wave equation is indeed important in describing physical processes related to propagation, further clarification is needed regarding its purpose in this model.
> > >
> > > The wave equation plays a significant role in atmospheric dynamics, and its connections to weather forecasting can be understood from several perspectives:
> > > - Atmospheric Waves: The wave equation directly describes atmospheric waves such as gravity waves and Rossby waves, which are crucial for understanding large-scale weather patterns and their propagation dynamics [1]. These waves are essential for explaining key phenomena in atmospheric circulation and energy transfer.
> > > - Global Circulation Models (GCMs): Many GCMs incorporate simplified or derived forms of the wave equation, adapted to the atmospheric context [2]. These models leverage the principles of fluid dynamics, with additional factors like the Coriolis effect and thermodynamic processes.
> > > - Data Assimilation: The wave equation also informs data assimilation techniques in weather prediction, where observational data is continuously integrated into models to refine predictions [3]. Understanding wave propagation is key to managing the integration of observational data and improving forecasts.
> > >
> > > In our study, we do not directly use the wave equation in its strongest bias, as a governing physical equation. **Instead, its high-level relationship with the advection equation serves as inspiration for our approach.**  The advection function estimates velocity from temporal derivatives, which rely heavily on fine temporal sampling.  This presents a challenge for us, as our input data is sampled hourly, with no higher temporal resolution available.  Such a high sampling ratio limits the accuracy of time-based velocity estimation.
> > >
> > > The wave equation, however, suggests an alternative: spatial derivatives can be used to infer temporal velocities if we assume that atmospheric waves govern the system.  **This insight allows us to leverage spatial information for velocity modeling, which significantly improves predictive performance.**  Our ablation study shows that incorporating spatial information improves results by over 10%, and we even find that using spatial information alone yields the best performance.  **In contrast, adding temporal information under high time-resolution constraints (1-hour sampling) decreases prediction accuracy.** These results underline the effectiveness of a spatially driven approach in addressing limitations posed by coarse temporal sampling.
> > >
> > > [2] Waves to Weather: Exploring the Limits of Predictability of Weather
> > >
> > > [3] Neural General Circulation Models for Weather and Climate
> > >
> > > [4] Local assimilation of wave model predictions for weather routing systems
> > >
> > > > If I understood correctly, the integrated wave equation (5) seems to show that the first-order derivative of the state ∂u/∂t(t0) is linked to the state gradients ∇u(t0), which motivates predicting the velocity field should as a function of the state gradients rather than the state itself. However, assuming that the wave equation holds, the involved spatial derivatives are of second order rather than first order, and are integrated over a past time interval rather than evaluated only at the current time t0. In my opinion, and if I understood correctly, equation (5) may not be entirely appropriate for predicting v(t0) as described.
> > >
> > > The reviewer is correct that in Equation (5), $\nabla u$ involves second-order derivatives, and computing $\frac{\partial u}{\partial t}$ also requires integration. **We believe these operations can be effectively learned by the neural network, as neural networks are capable of approximating both differentiation and integration.**
> > >
> > > The key insight from the wave equation is the discovery of a relationship between $\frac{\partial u}{\partial t}$ and $\nabla u$. Compared to $\frac{\Delta u}{\Delta t}$, which is used in ClimODE and derived from temporal differences, $\nabla u$ provides a more accurate estimate of $\frac{\partial u}{\partial t}$, thereby improving the prediction of $v(t_0)$. We will work to make this connection clearer in the revised version.

---

> ### Author Response · Authors · 2024-11-18
> **Response Continued**
>
> > This concern is compounded by the results of Figure 3, where the performance gap between the proposed method using the wave equation and other approximators is quite small. Couldn't it be that the performance gap comes from learning the initial velocity from a neural network conditioned on non-discretized state values instead of solving an inverse problem with discretized state derivatives, rather than from the wave equation-informed predictive structure of the neural network?
>
> The results in Figure 3 show that $f_v(\nabla u)$ achieves over a 10% improvement compared to $f_v\left(\frac{\Delta u}{\Delta t}\right)$, demonstrating that estimating $v$ using $\nabla u$ is more accurate than using $\frac{\Delta u}{\Delta t}$. This validates the effectiveness of the wave equation-based approach.
>
> **To further address your concern, we conducted similar experiments on ClimODE**, as shown in **Table A**. The results reinforce the advantages of the wave equation-informed predictive structure, providing additional evidence of its efficacy.
>
> **Table A.** Experiment of estimation of $v_0$ of ClimODE.
> |||RMSE|||ACC|||
> |-|-|-|-|-|-|-|-|
> |Variable|Lead Time|$f_v(\frac{\Delta u}{\Delta t})$ (ClimODE)|$f_v(\Delta u)$|$f_v(u, \Delta u)$|$f_v(\frac{\Delta u}{\Delta t})$ (ClimODE)|$f_v(\Delta u)$|$f_v(u, \Delta u)$|
> |z500|6|79.1|74.2|68.7|1.00|1.00|1.00|
> ||12|111.2|97.6|88.5|0.99|1.00|1.00|
> ||18|148.9|125.1|110.5|0.99|0.99|1.00|
> ||24|192.8|158.0|138.8|0.98|0.99|0.99|
> |t850|6|1.01|0.96|0.92|0.98|0.99|0.99|
> ||12|1.23|1.09|1.09|0.97|0.98|0.98|
> ||18|1.40|1.19|1.14|0.96|0.98|0.98|
> ||24|1.52|1.34|1.28|0.95|0.97|0.98|
> |t2m|6|1.13|1.00|0.97|0.97|0.98|0.99|
> ||12|1.36|1.13|1.07|0.97|0.98|0.98|
> ||18|1.54|1.22|1.19|0.96|0.98|0.98|
> ||24|1.75|1.28|1.18|0.96|0.98|0.98|
> |u10|6|1.15|1.09|1.10|0.95|0.97|0.98|
> ||12|1.41|1.26|1.23|0.94|0.96|0.97|
> ||18|1.68|1.41|1.37|0.93|0.95|0.96|
> ||24|1.83|1.61|1.53|0.90|0.94|0.95|
> |v10|6|1.21|1.12|1.08|0.95|0.97|0.97|
> ||12|1.47|1.30|1.29|0.94|0.96|0.97|
> ||18|1.65|1.47|1.39|0.93|0.95|0.96|
>
> > Finally, the qualitative argument about spatial resolution at the end of this section is not particularly convincing, as it compares spatial resolution to time resolution in a way that seems dimensionally inconsistent. The statement on line 243, **the spatial domain is nearly 100 times denser than the temporal domain.**
> needs further clarification, as its implications are unclear.
>
> Thank you for pointing this out. We acknowledge that the comparison between spatial and temporal resolution might appear dimensionally inconsistent. However, the key lies in their underlying periodicity. The temporal domain has an implicit period of 24 hours, while the spatial domain spans the entire Earth. This leads us to compare 1/24 (temporal) with 1/(32×64) (spatial).
>
> Since the units of time (hours) and space (kilometers) are not directly comparable, it is difficult to determine the "ideal" resolution between the two purely from theoretical arguments. To address this, we rely on experiments. In our Neural ODE, the velocity can be estimated temporally (using $\frac{\Delta u}{\Delta t}$) or spatially (using the wave function). Before conducting the experiments, we were uncertain which approach would work better. **However, we observed a clear improvement with spatial estimation, while adding temporal information often led to negative results.** This suggests that the temporal sampling frequency is severely insufficient, making time-based velocity estimation much less effective.
>
> To further support this point, **we supplement our experiments with precipitation forecasting tasks using the SEVIR dataset** [5], which features a temporal resolution of 5 minutes (significantly shorter than the 1-hour intervals in our primary experiments). **Table B** below demonstrates that when the temporal sampling is sufficiently dense, using $\frac{\partial u}{\partial t}$ instead of $\nabla u$ for velocity estimation becomes viable. The results indicate that dense temporal resolution mitigates the disadvantages of temporal derivative approximation.
>
> **Table B.** WeatherODE result on SEVIR dataset
> ||CSI $\uparrow$|CSI_pool4 $\uparrow$|CSI_pool16 $\uparrow$|
> |-|-|-|-|
> |WeatherODE ($f_v(\frac{\Delta u}{\Delta t})$)|0.1953|0.2031|0.2258|
> |WeatherODE ($f_v(\nabla u)$)|0.1926|0.2045|0.2290|
>
> (CSI is **Critical Success Index**)
>
> [5] SEVIR: A storm event imagery dataset for deep learning applications in radar and satellite meteorology.

---

> ### Author Response · Authors · 2024-11-18
> **Response Continued**
>
> > Neural ODEs are known to be computationally demanding, yet the paper does not address computational resources or the back-propagation method used to train the neural ODE. Since computational power can often be a limiting factor, it would be valuable to compare the computational times across the different architectures, particularly for the neural ODE versus other feedforward models.
>
> Deep learning with physics-informed models like WeatherODE, ClimODE, and NeuralGCM often have higher computational costs compared to purely data-driven models such as FourCastNet and Pangu. However, these models benefit from the ability to predict multiple time steps in a single inference pass, which makes them more versatile for extended forecasts. (**As WeatherODE$^{*}$ can generate predictions for all points within a 24-hour window in a single inference pass.**)
>
> In contrast, purely data-driven models typically require separate models for each lead time or rely on iterative rolling strategies, where outputs from one model serve as inputs for the next. This approach leads to repeated computations for multi-step forecasts. **As a result, for scenarios involving multiple consecutive lead times, physics-informed models can offer better computational efficiency by eliminating the overhead of repeated inference.**
>
> ## **Questions**
>
> > The WeatherODE* model appears to be an interesting research direction, offering flexibility in generating forecasts at different lead times. However, there is limited explanation as to why WeatherODE is capable of such flexibility. Could the authors clarify the connection between the following sentence:
> **by modeling the atmosphere as a physics-driven continuous process and designing a time-dependent source network to account for errors at each time step, WeatherODE can capture information across all intermediate time points**
> and the success of the 24-hour model WeatherODE*?
>
> Thank you for your question. Let me clarify. As mentioned earlier, the Neural ODE produces outputs $\hat{u} \in [N, K, H, W]$, where $N$ represents the desired lead time length. Each step along the $N$-dimension corresponds to predictions at specific future time points. For example, with $N = 24$, $\hat{u}[5,:,:,:]$ represents the forecast for the 6th hour, while $\hat{u}[14,:,:,:]$ corresponds to the 15th hour. These outputs are generated in a single pass through the Neural ODE. **We believe this design inherently captures temporal dependencies across adjacent time points during the ODE iterations of the "adjoint states", offering a significant advantage over purely data-driven models.**
>
> In contrast, models like Pangu or FourCastNet typically learn direct mappings between two discrete time points, such as data at $t$ and $t+6$ for a 6-hour forecast. **As a result, to predict different lead times, these models either require training separate models for each lead time or rely on rolling strategies with multiple models. This makes Neural ODE-based methods more naturally suited for capturing the continuous dynamics of the atmosphere.**
>
> Regarding the time-dependent source network, we apply it to the ODE outputs $\hat{u} \in [N, K, H, W]$ to model the potential effects of the source term. Instead of embedding the source term directly within the ODE iterations—which could amplify biases inherent to the physical process—we model the source influence separately using a 3D CNN that processes all $N$-dimensional outputs simultaneously. **The "time-dependent" aspect refers to our explicit consideration of temporal correlations across the $N$-dimension, leveraging the 3D CNN's ability to learn dependencies across time steps effectively.**
>
> ---
>
> We hope Reviewer g7RP finds these clarifications helpful and appreciates the reasoning behind our design choices. Thanks again for valuable feedback.

---

> > ### Comment · Reviewer_g7RP · 2024-11-22
> > **Answer to the authors**
> >
> > I appreciate the authors' thorough and thoughtful response, as well as their efforts to address my concerns by providing additional experimental results. These results strengthen the case for the practical advantages of WeatherODE and its improvements over the state of the art in the proposed experiments.
> >
> > Regarding the advection equation, I agree that, while it may not hold universally for all physical variables, it serves as a guiding principle encouraging mass conservation, which likely aids the neural network in learning effectively. I believe that the additional physical context provided by the authors in their response is valuable and should be incorporated into future versions of the paper. Similarly, the proposed clarity improvements for Section 3.1 are important and should enhance the paper's accessibility.
> >
> > On the topic of the wave equation, however, I remain unconvinced about the physical explanation motivating Section 3.2, as reviewers wKzX and o4jc also noted. In my view, the observed performance gains are more likely attributed to learning the initial velocity with a neural network conditioned on non-discretized state values, rather than solving an inverse problem using discretized state derivatives. This makes the wave equation justification feel somewhat superficial. Consequently, I suggest that future iterations of the paper provide more details on this learning approach, which appears to be the true driver of improvement, rather than emphasizing a physical motivation for including state gradients in the model.
> >
> > I thank the authors for their clarifications and responses, which have made aspects of the paper much clearer. However, I have decided to maintain my score. While the experimental results are strong, I find the technical contributions lack significant novelty and compared to ClimODE and physical insight. That said, I believe the authors are on a promising path. Enforcing physical biases is crucial for weather forecasting using machine learning, and I encourage the authors to continue exploring how neural architectures can be further improved in this direction to enhance both performance and theoretical grounding.

---

> > > ### Author Response · Authors · 2024-11-24
> > > **Response**
> > >
> > > Thank you for your continued engagement and for acknowledging the experimental strengths and clarity improvements we aim to incorporate into future versions. Regarding the wave equation and its role in our framework, we appreciate the opportunity to clarify further.
> > >
> > > The advection equation is indeed a specific case of the wave equation, as all solutions of the advection equation satisfy the wave equation. This can be demonstrated as follows:
> > >
> > > $
> > > \frac{\partial^2 u}{\partial t^2} - c^2 \frac{\partial^2 u}{\partial x^2} = \left( \frac{\partial u}{\partial t} + c \frac{\partial u}{\partial x} \right) \left( \frac{\partial u}{\partial t} - c \frac{\partial u}{\partial x} \right) u = 0.
> > > $
> > >
> > > Clearly, if $\frac{\partial u}{\partial t} + c \frac{\partial u}{\partial x} = 0$, then $u$ is also a solution of the wave equation. This relationship allows us to leverage the wave equation as a **weaker and more general bias** in our model. While the system itself adheres to the advection equation, introducing the wave equation provides additional flexibility by incorporating spatial gradients ($\nabla u$) to approximate temporal derivatives ($\frac{\partial u}{\partial t}$). This is especially advantageous when dealing with coarse temporal resolutions, as it mitigates the reliance on discretized time derivatives.
> > >
> > > We hope this explanation clarifies your concerns, particularly in the context of leveraging physical principles to guide model design. Thank you once again for your insightful questions and valuable feedback.

---

### Official Review · Reviewer_tB5a · 2024-10-28

**Soundness:** 2
**Presentation:** 2
**Contribution:** 2
**Rating:** 1
**Confidence:** 4

**Summary:**

This paper introduces a sandwich method whereby the atmospheric dynamics is represented by a simplified advection differentials and the components are separately learned (e.g., source, advection, initial condition) with a parameterized deep models, depending on whether they possess fast or slow convergence.

**Strengths:**

Building physics-informed network for chaotic systems, such as weather dynamics, is crucial and this paper offers an interesting ODE-based approach to tackle the problem.

**Weaknesses:**

Let me start off by saying that using an advection equation is only valid under __very simplified scenario__, and weather dynamics is definitely not the case at all. Elaborations to follow:

1. The paper assumes incompressible fluid, where fluid density remains static along the pressure level. This is a gross simplification and is unphysical as the weather dynamics is completely dependent on the fluid being compressible, to allow interesting processes to take place such as convection through buoyancy (cloud formation, precipitation, energy-water cycling), large-scale fluid motion (teleconnection), turbulence (boundary-layer interaction), etc. This very assumption, therefore, does not allow for many atmospheric phenomena, including but not limited to extreme events (ENSO, hurricanes, etc), or just the general circulation of the atmosphere. This is not including the paper's simplification of ignoring the spherical nature of the Earth that enables for a different set of dynamics enabled by e.g., coriolis force.

2. The paper also ignores many important conservation constraints found in the classical dynamical core, such as the conservation of mass, energy, and momentum.

3. As such, there are better approximation to the full Navier-Stokes equation, such as Shallow Water Equation or Quasi-Geostrophic flow, that attempts to capture some of the realism of the atmosphere, and is therefore a much better differentials than the advection equation.

Regardless, there are additional weaknesses that warrant a reject rating:

4. The limited number of variables used (5), coarse spatial resolution (5.625-degree vs 0.25; ~400x smaller horizontal resolution), and small forecasting lead-time (72-hour) are too unconvincing to test this Neural ODE formulation for real weather application. I suspect that given the gross oversimplification through the advection equation, the framework could not accurately evolve the full atmospheric state with significant vertical motion/interaction, and is therefore not useful in a short forecasting window, let alone over a longer rollout in the medium-weather and longer sub-seasonal scale.

5.  This is echoed by the result in Appendix E Table 9 is inferior to ClimaX at 3-day lead time, with error growth much larger than the other model (also lacking IFS baseline here to see how the rate of error propagation). The ACC for for u10, for instance, deteriorates from 0.93 to 0.80 even at 36-hour difference! The fundamental error in the assumptions (which ClimODE is too, and the authors should therefore use stronger baselines such as GraphCast) might contribute to this result.

Overall, the basic assumptions underpinning this ODE is over-simplistic and does not represent realistic atmosphere (e.g., no vertical motion, no conservation constraints). If the goal is to model a simplified flow, such as tracer dynamics where their evolution is passive e.g., does not have any feedback loop with other quantities, and local, then advection might be sufficient. But the atmosphere is definitely not the case. Also, the inferior results over long lead-time, weak baseline, and the limited scope of the problem setup, need more significant work.

**Questions:**

Similar as the weaknesses.

---

> ### Author Response · Authors · 2024-11-18
> **Response**
>
> We deeply appreciate Reviewer tB5a’s insightful and detailed feedback. The points raised demonstrate an exceptional understanding of the intricacies of weather forecasting and atmospheric modeling. Below, we provide detailed responses to each point and aim to clarify our approach further.
>
> > The paper assumes incompressible fluid, where fluid density remains static along the pressure level. This is a gross simplification and is unphysical as the weather dynamics is completely dependent on the fluid being compressible, to allow interesting processes to take place such as convection through buoyancy (cloud formation, precipitation, energy-water cycling), large-scale fluid motion (teleconnection), turbulence (boundary-layer interaction), etc.
>
> We acknowledge that assuming incompressibility is a simplification. However, our intention is not to fully replicate the complexity of weather dynamics but to explore a tractable approach for introducing physical insights into data-driven models. **Our assumption of mass conservation at a regional scale (approximately incompressible within our spatial patches) serves as a generalizable physical bias rather than a strict representation of atmospheric processes.**
>
> **We aim for a minimal yet effective incorporation of physical constraints, following Occam's Razor: "We consider it a good principle to explain the phenomena by the simplest hypothesis possible."** By simplifying the dynamics, we provide neural networks with just enough guidance to learn from the data while maintaining flexibility for capturing more nuanced, emergent patterns.
>
> We recognize the value of incorporating vertical dynamics and more complex interactions (e.g., buoyancy, turbulence). However, doing so requires significantly more computational resources, as further detailed below.
>
> > The paper also ignores many important conservation constraints found in the classical dynamical core, such as the conservation of mass, energy, and momentum.
>
> This is a valid point. While we agree that conservation laws such as energy and momentum are crucial for accurately modeling atmospheric dynamics, **we opt to focus on the simpler mass conservation principle as an initial step.** Mass conservation is one of the most fundamental constraints and provides a tractable starting point for neural ODE-based modeling, particularly given the resource constraints discussed below.
>
> In future works, **we aim to explore methods for integrating additional conservation constraints into our framework while balancing computational feasibility and model flexibility.**
>
> > As such, there are better approximations to the full Navier-Stokes equation, such as Shallow Water Equation or Quasi-Geostrophic flow, that attempts to capture some of the realism of the atmosphere, and is therefore a much better differential than the advection equation.
>
> We agree that models like the Shallow Water Equation or Quasi-Geostrophic flow offer better approximations of atmospheric dynamics compared to the advection equation. **However, our primary goal is to explore the potential of neural ODEs in weather forecasting, leveraging a simplified equation to provide flexibility for data-driven learning.** The advection equation, derived from mass conservation, offers a reasonable starting point as it captures essential transport dynamics while introducing minimal physical bias.
>
> **Our choice is not to suggest that the advection equation is a complete representation of atmospheric dynamics but rather to create a framework that can be scaled and adapted as computational resources and data availability improve.** As resources permit, incorporating more sophisticated equations like the Shallow Water Equation is a natural next step.

---

> ### Author Response · Authors · 2024-11-18
> **Response Continued**
>
> >  The limited number of variables used (5), coarse spatial resolution (5.625-degree vs 0.25; ~400x smaller horizontal resolution), and small forecasting lead-time (72-hour) are too unconvincing to test this Neural ODE formulation for real weather application.
>
> This is a fair concern, and we acknowledge the limitations of using only five variables and a 5.625° resolution. However, these choices were driven by the computational constraints faced by our research team. **Training a neural ODE model at 1.5° or 0.25° resolution requires substantial resources—potentially hundreds or thousands of GPUs—making it feasible only for major tech companies like Google, Meta, or NVIDIA.** For instance, Google's GraphCast achieves 0.25° resolution without neural ODEs, and even their neuralGCM model is limited to 1.4° resolution due to the computational demands.
>
> While our current setup is indeed coarse, we see this as a necessary first step for demonstrating the feasibility of neural ODEs in this context. **The positive results at 5.625° resolution suggest the potential for improved performance as we scale to finer resolutions, as supported by established scaling laws in machine learning.**
>
> To further support this point, we supplemente our experiments with precipitation forecasting tasks using the SEVIR dataset [1], which features a temporal resolution of 5 minutes (significantly shorter than the 1-hour intervals in our primary experiments). **Table A** below demonstrates that when the temporal sampling is sufficiently dense, using $\frac{\partial u}{\partial t}$ instead of $\nabla u$ for velocity estimation becomes viable. The results indicate that dense temporal resolution mitigates the disadvantages of temporal derivative approximation.
>
> **Table A.** WeatherODE result on SEVIR dataset
> ||CSI $\uparrow$|CSI_pool4 $\uparrow$|CSI_pool16 $\uparrow$|
> |-|-|-|-|
> |WeatherODE ($f_v(\frac{\Delta u}{\Delta t})$)|0.1953|0.2031|0.2258|
> |WeatherODE ($f_v(\nabla u)$)|0.1926|0.2045|0.2290|
>
> (CSI is **Critical Success Index**)
>
> [1] SEVIR: A storm event imagery dataset for deep learning applications in radar and satellite meteorology
>
> > This is echoed by the result in Appendix E Table 9, which is inferior to ClimaX at 3-day lead time, with error growth much larger than the other model (also lacking IFS baseline here to see how the rate of error propagation). The ACC for u10, for instance, deteriorates from 0.93 to 0.80 even at 36-hour difference! The fundamental error in the assumptions (which ClimODE is too, and the authors should therefore use stronger baselines such as GraphCast) might contribute to this result.
>
>
> We appreciate this observation and agree that our model shows limitations over longer lead times, with notable error growth. The absence of an IFS baseline in Table 9 was an oversight that we will address in future revisions. Comparing our model to stronger baselines like GraphCast would be ideal, but doing so at 5.625° resolution is challenging given the significantly finer resolution (0.25°) of GraphCast.
>
> **While our results highlight some limitations, they also point to the potential of neural ODEs for capturing atmospheric dynamics. For example, our model's RMSE for 24-hour forecasts is comparable to that of downsampled EC-IFS results at 5.625°.** These findings suggest that our approximation may not be as superficial as it seems and could achieve competitive results at finer resolutions.
>
> > If the goal is to model a simplified flow, such as tracer dynamics where their evolution is passive e.g., does not have any feedback loop with other quantities, and local, then advection might be sufficient. But the atmosphere is definitely not the case.
>
> We agree that the atmosphere involves complex feedback loops and interactions that go beyond what the advection equation can capture. **However, our goal is not to create a comprehensive physical model but to explore how simplified physical principles can guide data-driven learning.** While advection provides only a partial description of atmospheric processes, its simplicity offers neural networks the flexibility to learn additional patterns from data.
>
> As noted earlier, our approach reflects a tradeoff: **we incorporate minimal physical bias to guide learning while maintaining enough flexibility to adapt to data-driven insights.** This balance is critical for handling the inherent uncertainties in atmospheric dynamics, particularly given the chaotic nature of the weather system.
>
> ---
>
> We hope Reviewer tB5a appreciates the reasoning behind these design choices and acknowledges the constraints faced by researchers outside major institutions. We value the feedback and believe it will help refine and advance our work in future iterations. Thanks again for thoughtful comments.

---

> ### Comment · Reviewer_tB5a · 2024-11-19
> **Response**
>
> I appreciate the authors' response to my feedback. However, the inclusion of precipitation highlights a fundamental flaw in the approach—compressible fluid dynamics are essential for forecasting precipitation, as changes in fluid density across vertical levels (i.e., adiabatic processes) cannot be ignored. The assumption that the system is "approximately incompressible within spatial patches" is even more problematic, as precipitation and other weather phenomena rely on localized vertical motion, including microphysical processes. Resolving these small-scale processes remains a major challenge in climate modeling, often necessitating parameterization [1] (clarification: this is still an extremely challenging task even with the full primitive equation as the dynamical core; and adding more uncertainty to this already uncertain task through the use of overly simplified dynamics to the point of unphysical, and forcing the model to somehow learn this in a fully data-driven way is less helpful and may impede progress). While I understand the intent to simplify dynamics, the advection equation is inappropriate for this case, and I recommend the authors to consider NeuralGCM [2], which employs a more appropriate simplified dynamical core (e.g., the Shallow Water Equation) that better approximates the Navier-Stokes equations. In short, the use of advection equation is fatally incorrect, and in light of the rebuttal and to avoid this error from appearing in future works, I reduce my score to reflect this.
>
> __References:__
>
> [1] Rasp, Stephan, Michael S. Pritchard, and Pierre Gentine. "Deep learning to represent subgrid processes in climate models." Proceedings of the national academy of sciences 115.39 (2018): 9684-9689.
>
> [2] Kochkov, Dmitrii, et al. "Neural general circulation models for weather and climate." Nature 632.8027 (2024): 1060-1066.

---

> > ### Author Response · Authors · 2024-11-20
> > **Response**
> >
> > We believe we have a philosophical disagreement with Reviewer tB5a regarding our approach and research direction. We would like to clarify our stance by presenting four questions, ranging from simple to complex, to articulate our points of contention.
> >
> > - **Q1:** If an assumption that underpins a mathematical proof does not hold, is the work fundamentally flawed and deserving of rejection?
> > - **A1:** Yes, we agree in this scenario. It is crucial for the author to verify the assumptions to resolve the conflict.
> >
> > ---
> >
> > - **Q2:** If the assumption doesn't hold for a numerical problem requiring modeling and calculation from a PDE/ODE or pure mathematical perspective, is the work incorrect and should it be strongly rejected?
> > - **A2:** It depends. The decision involves balancing no assumptions, incorrect assumptions, and correct assumptions. While real-world constraints (e.g., computational power) might prevent the use of the correct assumption, an incorrect assumption could still yield better results compared to having no assumptions. Typically, the correct assumption leads to better outcomes, if feasible.
> >
> > ---
> >
> > - **Q3:** If the assumption doesn't hold for a bias introduced in a deep learning work (AI4science), is the work flawed and should it be strongly rejected?
> > - **A3:** No, we argue this is not the case. Consider the well-known example of Convolutional Neural Networks (CNNs). CNNs revolutionized AI and remain widely used, despite the primary assumption of shift-invariance not holding in most cases, such as visual or NLP applications. This assumption aids in optimization but doesn’t strictly confine the algorithm. The fundamental concept of universal approximation still applies, allowing models to learn complex functions irrespective of introduced biases. Similarly, our approach with the advection equation, acknowledging it doesn't strictly hold, shows improvement with marginal benefits compared to the no-assumption approach. If we check works like Pangu, GraphCast, Fuxi, or ClimaX, they don’t rely on local interaction/compressibility as Reviewer tB5a suggests, nor do they assume adherence to a single ODE. Yet, these approaches achieve successful results with minimal assumptions.
> >
> > ---
> >
> > - **Q4:** Is pursuing the path with the most complex and ostensibly correct assumptions superior to using minimal or even incorrect assumptions in neural networks? Should approaches with fewer assumptions be strongly rejected?
> > - **A4:** No, we argue this is not the case. We had extensively reviewed the NeuralGCM approach, which integrates numerous ODE functions used in numerical weather prediction (NWP). While Reviewer tB5a suggests this is the right path, we posit the opposite. Our approach seeks a balance between minimal physical biases and the flexibility of machine learning, with a stronger inclination towards a data-driven direction. We aim for a model less biased than NeuralGCM. This debate is not new and mirrors the clash between Geoffrey Hinton and his supervisor Christopher Longuet-Higgins, who thought symbolic AI with complex assumptions was favored. Hinton believed in a different path, accelerating the AI era. We advocate for fewer assumptions, allowing models to learn primarily from data. While the scarcity and complexity of weather data make complete bias removal challenging, incorporating all PDE biases as in GCM is not our goal. Although the optimal path in weather forecasting remains open, successes in fields like CV and NLP, along with examples from weather forecasting such as Pangu, GraphCast, and Fuxi, validate the effectiveness of approaches with minimal bias. Therefore, pursuing the most complex set of assumptions may not be the ideal approach for neural networks.

---

### Official Review · Reviewer_wKzX · 2024-11-04

**Soundness:** 2
**Presentation:** 3
**Contribution:** 2
**Rating:** 3
**Confidence:** 4

**Summary:**

The paper is related to weather modeling on a 5.625° resolution using ERA-5 data as target. The solution is based on the NeuralODE model with several new features inside. A CNN-ViT-CNN sandwich architecture is proposed to model the right term of the ODE. The authors claim a strong improvement in simulation quality over the alternative approaches. Such an improvement is attributed to new scheme of velocity derivatives estimation based on the wave equation and to the CNN-based approximation of a source term in the advection equation

**Strengths:**

* An interesting wave-equation inspired model to capture time derivative of velocity
* Comparison with several baselines showing better results of a proposed model
* Ability to interpolate in time and produce a continuous dynamics
* Ablation studies for the velocity derivative approximation and model architecture

**Weaknesses:**

* The CNN used in variable space scale loses part of its physical meaning
* The wave equation doesn’t describe the atmospheric dynamics really well so the derivative approximation is not governed by physics
* The model has low practical applicability
Probably an interesting experiment will be to use the model in the autoregressive mode and to assess its quality for longer forecast horizons. The same can be done for regional models to assess the borders influence.

**Questions:**

* How the model will perform on poles where the horizontal resolution is very different from the equator?
* Do your CNN implementation consider the discontinuity in latitudes (360=0)?
* The border influences much the model during regional forecasts. How do you handle this problem? (if wind velocity is 100 km/h, in 24 hours it will travel 2400 km)
* How your model can be used in practice? Consider that ERA-5 dataset is not available operationally.

---

> ### Author Response · Authors · 2024-11-18
> **Response**
>
> We sincerely thank Reviewer wKzX for their thoughtful feedback. The questions posed are both intriguing and align with previous studies we've conducted but did not include in the paper. Below, we address each point raised in detail.
>
> ## **Weakness**
>
> > The CNN used in variable space scale loses part of its physical meaning.
>
> We acknowledge this point, and this question is closely related to what we believe is the path to pursue in weather forecasting. Our approach aims to strike a balance between incorporating minimal physical biases and leveraging the flexibility of machine learning. But our balance leans more toward a data-driven direction.**Ideally, a completely data-driven model could fully capture atmospheric dynamics without introducing any biases. However, this would require vast amounts of data and computational resources, especially to model rare weather events.**
>
> To address these limitations, we incorporate simple and generalizable physical principles, such as advection, which introduce weak biases to guide the neural network. This also adheres to **Occam’s Razor: "We consider it a good principle to explain the phenomena by the simplest hypothesis possible."** By combining these weak biases with data-driven learning, we aim to provide enough guidance to learn robust representations while maintaining flexibility to model complex patterns beyond the physical assumptions.
>
> > The wave equation doesn’t describe the atmospheric dynamics really well so the derivative approximation is not governed by physics
>
> We agree that the wave equation is a simplification and does not fully describe atmospheric dynamics. However, its high-level relationship to the advection equation inspired our approach. **Specifically, the wave equation highlights how spatial gradients can inform temporal derivatives, enabling us to overcome limitations posed by the low temporal resolution (1-hour intervals) of our input data.**
>
> Our ablation studies demonstrate the effectiveness of incorporating spatial information, with results showing a significant improvement (10%) when modeling velocity using spatial derivatives rather than temporal differences. While the wave equation itself is not directly integrated into the model, it provides a conceptual foundation for improving derivative approximation.
>
> To further support this point, we supplement our experiments with precipitation forecasting tasks using the SEVIR dataset [1], which features a temporal resolution of 5 minutes (significantly shorter than the 1-hour intervals in our primary experiments). **Table A** below demonstrates that when the temporal sampling is sufficiently dense, using $\frac{\partial u}{\partial t}$ instead of $\nabla u$ for velocity estimation becomes viable. The results indicate that dense temporal resolution mitigates the disadvantages of temporal derivative approximation.
>
> **Table A.** WeatherODE result on SEVIR dataset
> ||CSI $\uparrow$|CSI_pool4 $\uparrow$|CSI_pool16 $\uparrow$|
> |-|-|-|-|
> |WeatherODE ($f_v(\frac{\Delta u}{\Delta t})$)|0.1953|0.2031|0.2258|
> |WeatherODE ($f_v(\nabla u)$)|0.1926|0.2045|0.2290|
> (CSI is **Critical Success Index**)
>
> [1] SEVIR: A storm event imagery dataset for deep learning applications in radar and satellite meteorology.
>
> Additionally, we conducted experiments on ERA5 data to analyze the impact of increasing the temporal interval $\Delta t$ when estimating $\frac{\Delta u}{\Delta t}$ for initial velocity $v(t_0)$. **Table B** demonstrates that as $\Delta t$ increases, the accuracy of $v(t_0)$ estimation decreases significantly, sometimes leading to numerical instability in the ODE. These results highlight the challenges of relying on temporal differences with coarse sampling intervals.
>
> **Table B.**  z500 result of the time interval $\Delta t$ for estimating $\frac{\Delta u}{\Delta t}$ to calculate $v(t_0)$. NaN indicates numerical instability. Full results in Table 12 of the paper.
> |||RMSE $\downarrow$|||||ACC $\uparrow$|||||
> |-|-|-|-|-|-|-|-|-|-|-|-|
> |Variable|Hours|$\Delta t=1$|$\Delta t=2$|$\Delta t=3$|$\Delta t=12$|$\Delta t=24$|$\Delta t=1$|$\Delta t=2$|$\Delta t=3$|$\Delta t=12$|$\Delta t=24$|
> |z500|6|71.0|86.8|88.8|107.8|140.5|1.00|1.00|1.00|0.99|0.98|
> ||12|100.6|118.5|128.7|164.3|NaN|0.99|0.99|0.99|0.98|NaN|
> ||18|134.0|157.3|161.3|NaN|NaN|0.99|0.99|0.99|NaN|NaN|
> ||24|172.8|NaN|NaN|NaN|NaN|0.98|NaN|NaN|NaN|NaN|

---

> > ### Author Response · Authors · 2024-11-18
> > **Response Continued**
> >
> > > The model has low practical applicability Probably an interesting experiment will be to use the model in the autoregressive mode and to assess its quality for longer forecast horizons. The same can be done for regional models to assess the borders influence.
> >
> > We have experimented with autoregressive mode and found that our neural ODE-based model performs worse than direct forecasting for long lead times. The results, shown in **Table C**, demonstrate the higher RMSE of rolling forecasts using a 6-hour WeatherODE compared to a 12-hour WeatherODE model. **We hypothesize that this is due to the model's higher sensitivity to initial conditions compared to purely data-driven approaches like Pangu.** Since our model incorporates physical biases, it is more reliant on accurate initial inputs. When these inputs deviate from reality, the sensitivity can amplify errors over time, resulting in suboptimal autoregressive performance.
> >
> > We believe this tradeoff arises from our approach’s emphasis on maintaining physical consistency, which increases bias but reduces variance. In contrast, purely data-driven models exhibit higher variance but are less sensitive to initial conditions. Future work will explore strategies to mitigate this sensitivity while retaining the advantages of physical biases.
> >
> > **Table C.** RMSE$\downarrow$ comparing the 12-hour WeatherODE model with the 6-hour WeatherODE model in a rolling forecast mode. For rolling forecasts, the 6-hour model is applied twice to achieve a 12-hour lead time.
> > ||z500|t850|t2m|u10|v10|
> > |-|-|-|-|-|-|
> > WetherODE (12h)|80.0|0.87|0.88|1.00|1.04|
> > WeatherODE (6h rolling twice)|105.6|1.10|1.12|1.45|1.48|
> >
> > ## **Questions**
> >
> > > How the model will perform on poles where the horizontal resolution is very different from the equator?
> >
> > We evaluate the model on polar regions and found that the performance is comparable to equatorial regions. **Specifically, the mean squared error (MSE) for test set samples in the poles (latitudes $[81.5626, 90]$ and $[-90, -81.5626]$) is 0.41, while the MSE in the equatorial region (latitudes $[-8.4375, 8.4375]$) is 0.36. This indicates similar performance across these regions, which suggests the model has effectively learned the differences between the equator and the poles through latitude-weighted coefficients $\alpha(\cdot)$.**However, further improvements on poles are possible. Although we followed KARINA[2]'s approach, no noticeable improvement was achieved.
> >
> > Additionally, **we visualize three representative samples in this link: https://postimg.cc/k2TYRns5.** These samples further confirm the model's robustness across different latitudinal resolutions.
> >
> > > Do your CNN implementation consider the discontinuity in latitudes (360=0)?
> >
> > This is a valid concern. While it is possible to design CNNs that address this discontinuity, such as by using circular padding (as demonstrated in KARINA [2]), **we found that applying such methods to our architecture provided no noticeable improvement over a simpler CNN design. For simplicity and efficiency, we chose not to include this modification.**
> >
> > We appreciate the reviewer highlighting this issue, as it is indeed important for global forecasting. We will reference relevant works in our revised version and consider further investigation into this design aspect in future work.
> >
> > [2] KARINA: An Efficient Deep Learning Model for Global Weather Forecast
> >
> > > The border influences much the model during regional forecasts. How do you handle this problem? (if wind velocity is 100 km/h, in 24 hours it will travel 2400 km)
> >
> > This is a limitation of our current implementation. For simplicity, we assume that the regional model operates in isolation without interactions from neighboring regions. We agree that a more realistic approach would involve a local-global framework, where fine-resolution local models are integrated with coarser-resolution global models to better handle boundary conditions. NVIDIA’s StormCast [2] exemplifies such a hybrid approach.
> >
> > **However, our study follows ClimODE's simplified setup to ensure direct comparability of results. Future extensions could incorporate local-global modeling to address boundary interactions more effectively.**
> >
> > [2] Kilometer-Scale Convection Allowing Model Emulation using Generative Diffusion Modeling

---

> > > ### Author Response · Authors · 2024-11-18
> > > **Response Continued**
> > >
> > > > How your model can be used in practice? Consider that ERA-5 dataset is not available operationally.
> > >
> > > To apply our model operationally, **we suggest using initial fields from ECMWF's services, which are available for purchase.** Even without access to ECMWF data, researchers can utilize ECMWF’s freely available forecast data as an alternative to the initial fields. Additionally, free initial fields provided by the U.S. GFS weather service can also be used.
> > >
> > > To adapt the model to GFS data, **we recommend fine-tuning the pretrained model for 10–20 epochs or adding a parameter-efficient fine-tuning (PEFT) layer while keeping other parameters frozen.** These adaptations should allow the model to operate effectively with GFS initial fields, making it practical for real-world applications.
> > >
> > > We hope Reviewer wKzX finds these clarifications helpful and appreciates the reasoning behind our design choices. Thanks again for valuable feedback.

---

> > > > ### Comment · Area_Chair_kZNh · 2024-11-26
> > > >
> > > > Dear reviewer,
> > > >
> > > > Please make to sure to read, at least acknowledge, and possibly further discuss the authors' responses to your comments. Update or maintain your score as you see fit.
> > > >
> > > > The AC.

---

> > > > > ### Comment · Reviewer_wKzX · 2024-11-27
> > > > > **I decided to keep the rating**
> > > > >
> > > > > Dear Authors,
> > > > >
> > > > > Thank you for going through all my questions. However I find not all of my concerns addressed (some of the answers are mostly philosophic) and also by going through all the discussion with all the reviewers I understand that there are more problems in the proposed work. That's why I decided to keep the same rating.
> > > > >
> > > > > I would advise probably the authors to make better investigation on practical usability of the obtained results (for example to train on GFS with 1/4 degree resolution) and if it works well to submit the work to some geophysical journal.

---

### Comment · Area_Chair_kZNh · 2024-11-26

Dear all,

The deadline for the authors-reviewers phase is approaching (December 2).

@For reviewers, please read, acknowledge and possibly further discuss the authors' responses to your comments. While decisions do not need to be made at this stage, please make sure to reevaluate your score in light of the authors' responses and of the discussion.

- You can increase your score if you feel that the authors have addressed your concerns and the paper is now stronger.
- You can decrease your score if you have new concerns that have not been addressed by the authors.
- You can keep your score if you feel that the authors have not addressed your concerns or that remaining concerns are critical.

Importantly, you are not expected to update your score. Nevertheless, to reach fair and informed decisions, you should make sure that your score reflects the quality of the paper as you see it now. Your review (either positive or negative) should be based on factual arguments rather than opinions. In particular, if the authors have successfully answered most of your initial concerns, your score should reflect this, as it otherwise means that your initial score was not entirely grounded by the arguments you provided in your review. Ponder whether the paper makes valuable scientific contributions from which the ICLR community could benefit, over subjective preferences or unreasonable expectations.

@For authors, please respond to remaining concerns and questions raised by the reviewers. Make sure to provide short and clear answers. If needed, you can also update the PDF of the paper to reflect changes in the text. Please note however that reviewers are not expected to re-review the paper, so your response should ideally be self-contained.

The AC.

---

### Author Response · Authors · 2024-11-27
**Seek suggestions from this knowledgeable and willing panel of engaged reviewers**

It appears that many of the reviewers for our submission are experts in Numerical Weather Prediction or Weather forecasting in general and have thoroughly examined our paper. They actively engaged in discussions, making them the ideal reviewers we could have hoped for. However, the outcome was devastating, with 3 out of 5 reviewers decreasing their original scores. In my several years of submitting, this has been a truly unprecedented experience. Additionally, the criticisms we received were unexpected and rarely focused on the improvements or model design, aspects on which we spent most of our time, working under flexible assumptions.

And after reviewing other submissions for ICLR25, it seems that all physics-informed weather forecasting work struggled to achieve satisfactory results. These efforts have also faced significant scrutiny due to the rigidity of their physical assumptions and the applicability of their equations.

As a result, we have a sincere question: Would a purely machine learning-based weather forecasting approach, without any physical equations, be more readily accepted by the community? It appears that having no assumptions might be better than having non-strict assumptions. We noticed that almost all weather-related papers would be rejected, so we might not get a chance to seek advice from successful works in this domain at the conference. Our decision to pursue this path was partly influenced by an oral presentation at the last ICLR conference titled 'Climate and Weather Forecasting with Physics-Informed Neural ODEs.' We believed it would be accepted as a baseline for improvement, but evidently, its basic assumptions are not acceptable, at least in the eyes of my reviewers.

If possible, could you provide us with some guidance? Despite our poor results, we still aspire to conduct research in this area. However, if the community is not receptive or finds this angle of work difficult to accept, we may consider making some adjustments.

---

> ### Comment · Reviewer_dcuz · 2024-11-27
>
> I sympathize with the authors regarding the unfortunate state of peer-review conducted at top ML conferences such as ICLR. I too have had a not-so-ideal experience this year where reviewers have been harsh towards my paper for no good reason. It is unfortunate that the reviews from non-serious reviewers (who barely read or try to understand the work) carries the same weight as serious reviewers (who put geniune effort in reading the paper).
>
> Regarding reducing the score, yes I am one of the reviewers who reduced the score. But I maintain my stance. I have been objective about my evaluation. I gave it a solid 8 after first reading. But, as is the case usually - doing good science takes time - time to understand new ideas and appreciate them - time to connect them with the big picture, and so on. The review format of these conferences does not allow enough time to read and understand these paper thoroughly - more so when one's schedule is already packed. After understanding your work more properly the second time, more limitations of the work popped up to me. Yet, because I like the overall idea, I have maintained a final decision to 'accept' the manuscript so as to not be unfair.
>
> You have a nice manuscript and a nice idea which has the potential to be a good contributions to the field. However, there are mutiple deficiencies in your approach as well which would prevent longer-term rollouts while being efficient - this ultimately conflicts with the usefulness of the approach. I think my concerns are grounded in scientific principles which, at the end of the day, are needed for a robust experimental design. My background is both in weather&climate dynamics and machine learning so it is important that the work makes sense to me both from the physical science and data perspective.
>
> If you can find solid grounding to nullify the concerns I make, I would be happy to reconsider my score. Until then, it is my responsibility to provide as fair of an evaluation as I can to the manuscripts assigned to me. At this point I don't see any. Conducting ML experiments is relatively easy, providing rigorous justification as to why they work is tough. Best of luck for the final decision.

---

> > ### Author Response · Authors · 2024-11-27
> >
> > Thank you so much for the prompt response and thorough review. We greatly appreciate your insightful suggestions and opinions, even though we struggled to provide strong explanations for some points. It seems our work will not advance this time. However, we hope the reviewers can be lenient towards other weather forecasting submissions at this conference within the following few days, focusing more on their contributions to the community. We greatly value and welcome additional research to guide us in this niche field. Aside from NeuralGCM and ClimODE, we found few recognized physically-informed weather forecasting studies.  We believe even ClimODE might face challenges this year due to its advection equation assumption. NeuralGCM is excellent but requires significant computational resources, making it hard to follow and compete with. If LLM works are all required to surpass ChatGPT, few will stand out, and we'll see fewer innovative ideas. Once again, thanks to all the reviewers for actively engaging in the discussion and selflessly contributing reviews for the benefit of the community.

---

> > > ### Comment · Reviewer_tB5a · 2024-11-27
> > >
> > > I do sympathize with the authors, and while I see the point of trying to simplify the problem and incorporate minimal physics as an inductive bias to better constrain data-driven hybrid approach, I find the method unconvincing and provide suggestions below:
> > >
> > > 1. Although it is true that the primitive equation (PE) is more complex than the advection equation to represent a dynamical core, the former can still be reasonably integrated in time given your spatiotemporal resolution. If you need a more performant core, there are several linearized approximation to PE that is more suitable, and starting from there is better not just because they are, at present, one of our better approximation to the atmosphere + weather dynamics, but also because it serves as a good abstraction to actually solve the hard part: physics. The pure data-driven approach appears to be more palatable since it might be a more radical idea, but doing a hybrid approach, I agree, is more difficult because as in a causal inference world, what you do not include might be more important than what you actually include (e.g., removing compressibility, etc is a serious concern). With that said, you can use NeuralGCM dycore and use your fast-slow VIT sandwich approach to better capture the physics in a coarse resolution setting (which to my current understanding, they still use a simpler FFN module).
> > >
> > > 2. Elaborating on point 1, I do want to encourage the authors to continue working on this subfield, precisely because it is difficult and the stakes are high that the reviewers tend to be harsher and skeptical. As I see it now, there are three ongoing branches in this field: full data driven, hybrid modeling, and pure physics. Your work might fall in the second category, and your use of weaker physical prior than the traditionally accepted PE, might cause several concerns among the reviewers. My reject rating signals this to prevent other works from going deep into a potentially unproductive (though not incorrect) rabbit hole. In short, it is better to get the dycore to an acceptable level and solve the harder part: unresolved physics with ML/DL. Because ultimately, if the domain scientists are not accepting of the core assumptions, it might be less helpful to drive operationalization and integration for e.g., real-world decision / policy making.
> > >
> > > I do encourage the authors to continue working on this important yet far from being solved problem. It is precisely because the stakes are high and we could not afford to waste a few more years building up a literature with a shaky foundation (where the alternative is not orders-of-magnitude more computationally intensive as the authors claim), that ultimately justifies my reject rating for the paper in its current form.

---

> > > > ### Author Response · Authors · 2024-11-28
> > > >
> > > > We sincerely appreciate your suggestions and thank you for taking the time to review our work.

---

### Meta-Review · Area_Chair_kZNh · 2024-12-20

**Metareview:**

The reviewers recommend rejection (3-1-5-6-3). The paper presents a hybrid approach combining a Neural ODE with physical equations for weather forecasting. The approach is reasonably well-motivated, but the results are unconvincing compared to the state of the art (small number of physical variables, coarse spatial resolution, short forecasting lead time). The author-reviewer discussion has been constructive and has led to a number of clarifications. However, the discussion has also led several reviewers to decrease their scores. The main concern raised by the reviewers is the core assumption of the approach, which relies on an advection equation and assumes incompressible fluid dynamics. This assumption is fundamentally limiting for weather dynamics. The authors argue that this assumption only serves as a guiding principle to showcase the benefits of the architecture, but reviewers have not been convinced by this argument. Following the reviewers' opinions, I recommend rejection. I encourage the authors to reflect on the reviewers' comments and to revisit the perspective, goals, and assumptions of the approach before resubmitting to a future conference.

**Additional Comments On Reviewer Discussion:**

The discussion has also led several reviewers to decrease their scores.

---

### Decision · Program_Chairs · 2025-01-22

Reject